# Versatile and sensitive detection of mono- and poly(ADP-ribosyl)ation reveals XRCC1-dependent remodelling of PARP1 signalling

Helen Dauben [1], Mihaela Mihaljević[1], Andreas Kolvenbach [1], Maria Dilia Palumbieri [1], Chrysi Kapsali [2], Ina Huppertz [2,3] & Ivan Matić [1,3] ✉

ADP-ribosylation has long been recognised as a key regulator of essential signalling pathways, including the DNA damage response. However, only recent and ongoing technological advances are beginning to make it possible to investigate its distinct forms with molecular precision. Here, we design a 'mono-ADP-ribosylation blocking' strategy to develop sensitive, modular antibodies with high specificity for poly(ADP-ribosyl)ation. During peptide antigen generation, we identify a distinctive mass spectrometric signature that enables accurate mapping of poly(ADP-ribosyl)ation sites and helps prevent site mislocalization. Moreover, we affinity-mature mono-ADP-ribosylation and histone H3 site-specific antibodies. These tools reveal that, upon DNA damage, XRCC1 deficiency dramatically elevates the mono-ADP-ribosylation wave of PARP1 signalling, in addition to increasing poly(ADP-ribosyl)ation. This PARP1 hyperactivation leads to an increase in an unconventional form of ubiquitylation, recently shown to directly target mono-ADP-ribose in the DNA damage response and other signalling pathways. Consequently, XRCC1 loss enhances the recruitment of RNF114, the reader of this composite modification, to DNA lesions. These findings establish mono-ADP-ribosylation – and its ester-linked ubiquitylation – as key modifications induced by XRCC1 deficiency during DNA damage, revealed using tools we developed for precise and sensitive ADP-ribosylation detection.

ADP-ribosylation (ADPr) is a versatile post-translational modification (PTM) that regulates numerous cellular processes, including DNA repair, transcription, chromatin remodelling, signalling and cell death[1–3]. It involves the enzymatic transfer of ADP-ribose units from nicotinamide adenine dinucleotide (NAD+) to target molecules, mediated primarily by members of the ADP-ribosyltransferase family[2]. This dynamic modification can occur as mono-ADPr, linear or branched poly-ADPr, and is reversed by ADP-ribosylhydrolases[4]. First identified in the 1960s[5], ADPr has since emerged as a critical player in cellular stress responses. Its dysregulation is implicated in various

diseases, including cancer and neurodegeneration, making it an attractive target for therapeutic intervention[6].

The poly(ADP-ribose) polymerase (PARP) family of ADP-ribosyltransferases comprises 17 members in humans[2]. PARP1 is activated in response to DNA strand breaks, coordinating the recruitment of DNA repair machinery and chromatin relaxation[7,8]. As a central scaffolding protein in single-strand break repair (SSBR), XRCC1 is recruited to sites of DNA damage through its interaction with poly-ADPr synthesised by PARP1/PARP2[9], enabling the formation of repair complexes. Loss of XRCC1 has been shown to cause

[1]Research Group of Proteomics and ADP-Ribosylation Signaling, Max Planck Institute for Biology of Ageing, Cologne, Germany. [2]Research Group of RNA-Binding Proteins in Metabolism and Ageing, Max Planck Institute for Biology of Ageing, Cologne, Germany. [3]Cologne Excellence Cluster for Aging and Aging-Associated Diseases (CECAD), University of Cologne, Cologne, Germany. ✉e-mail: imatic@age.mpg.de

elevated poly-ADPr levels through PARP1 hyperactivation, which has been linked to neurodegeneration[10,11]. By restraining PARP1 recruitment and catalytic activity during base excision repair (BER), XRCC1 prevents persistent PARP1 binding to BER intermediates. Recent work has further clarified how XRCC1 loss influences the temporal dynamics of PARP1 signalling. In particular, Demin et al.[10] demonstrated that in MMS-treated XRCC1-deficient cells, early PARP1 hyperactivation drives rapid NAD$^+$ consumption, suppressing PARP1 auto-modification at later time points. These findings indicate that XRCC1 not only restrains PARP1 activation but also preserves NAD$^+$ availability necessary for sustained ADPr responses. Moreover, sustained PARP1 activity induced by XRCC1 deficiency

has been shown to inhibit transcriptional recovery after DNA damage[12].

ADPr is difficult to study due to its multi-layered complexity. Its biological complexity stems from the diversity of its enzymes, substrates, and cellular contexts involved, with conjugation chemistries ranging from O-glycosidic (serine and tyrosine), S-glycosidic (cysteine), N-glycosidic (arginine) to ester linkages on aspartate and glutamate. Some conjugation chemistries have remained elusive for decades. A prime example of this is serine ADPr, discovered less than ten years ago despite its abundance[13–15]. The structural complexity of ADPr arises from its versatility to form both mono-ADPr and polymeric chains. This biological, chemical and structural complexity has

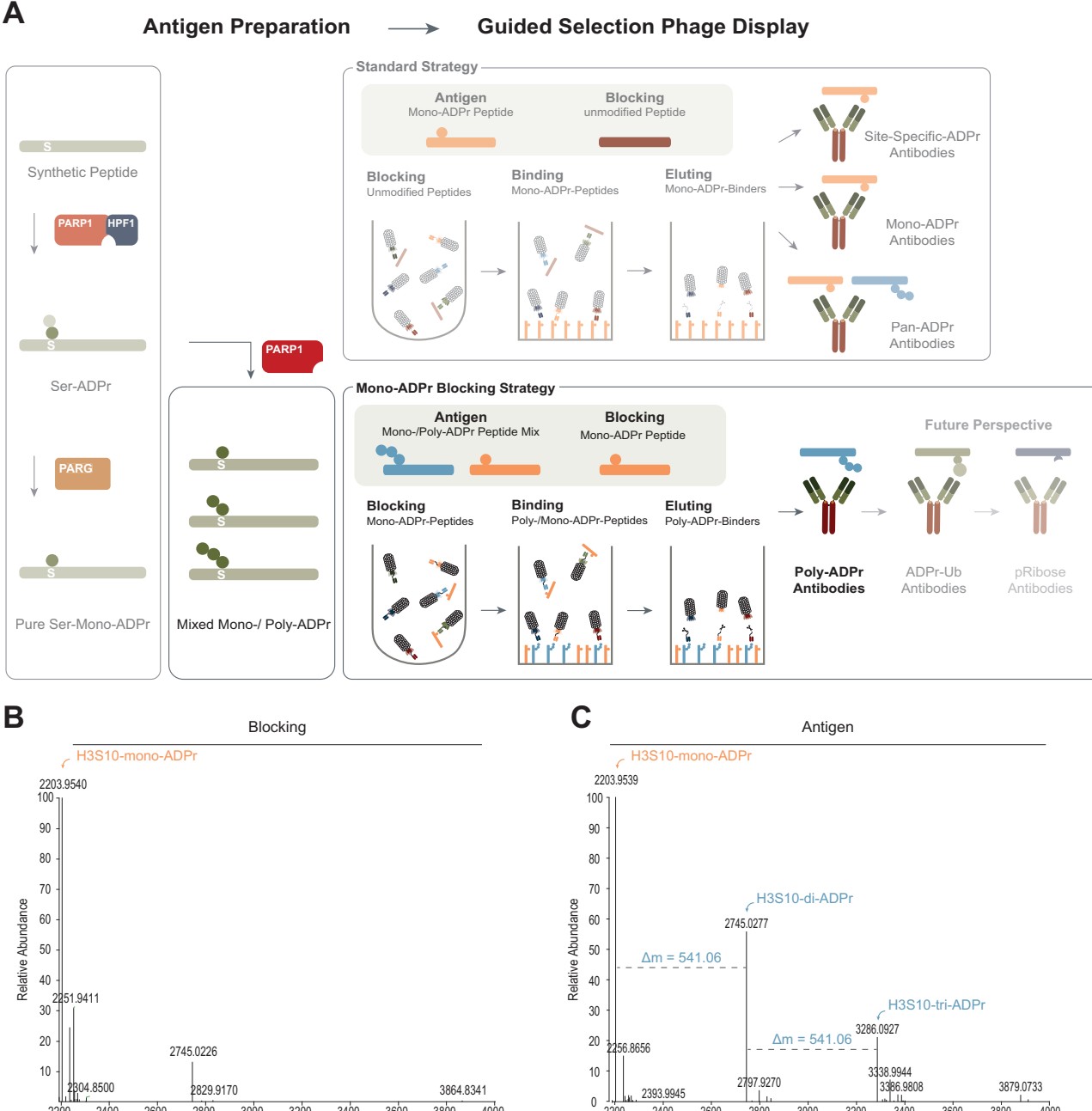

**Fig. 1 | A 'mono-ADP-ribosylation blocking' strategy. A** Schematic illustration of the mono-ADPr blocking strategy to generate anti-poly-ADPr antibodies. Illustrations generated in Adobe Illustrator. **B** Deconvoluted MS spectra of Histone H3 peptide after enzymatic Ser-mono-ADPr reaction used for blocking. $N = 1$.

**C** Deconvoluted MS spectra of Histone H3 peptide after enzymatic Ser-poly-ADPr reaction used as Antigen. As depicted, the in vitro reaction produces a shift of 541.06 Da (mass of ADPr = 541.06 Da) per attached ADPr unit, resulting in a mixture of mono- and different poly-ADPr species. $N = 1$.

historically hindered the study of ADPr by limiting the development of tools. The past decade has seen a rapid expansion of methodologies and tools that have advanced the study of ADPr[16], but a comprehensive toolbox for ADPr remains in development.

Our laboratory has contributed to extending the ADPr toolkit by converting our discoveries of serine ADPr by HPF1/PARP1[14,15] into a programmable technology for generating ADPr peptides and, subsequently, anti-ADPr antibodies and ADP-ribosylated nucleosomes[17–21]. This includes the first site-specific ADPr antibodies[21], antibodies for mono-ADPr detection in diverse signalling pathways[17,18,21–30] and in the context of ADP-ribosyl-ubiquitylation[31–36]. Serine ADPr technology is crucial because it not only generates antigens for phage display-based clone selection, but also enables in-depth validation and antibody characterisation. For example, the resistance of serine ADPr to hydroxylamine recently demonstrated the specificity of a reagent we developed for detecting ADP-ribosyl-ubiquitylation[33].

Phage display rapidly identifies multiple antigen binders in non-standard formats, but converting them to immunoglobulins through cloning is impractical and time-consuming. Fortunately, the SpyTag/ SpyCatcher[37] protein ligation technology now rapidly produces various antibody formats, including rabbit, mouse and human synthetic immunoglobulins, along with site-specific labelling with horseradish peroxidase (HRP), biotin, and other tags[38]. Immunoblotting with bivalent antibodies carrying three HRP molecules offers a more sensitive, simpler and faster alternative to standard protocols based on secondary antibodies[38], greatly improving ADPr detection[17,18]. Beyond synthetic antibodies, we recently applied the SpyTag system to engineer a protein domain for enriching and detecting cellular ADP-ribosyl-ubiquitylation[33].

While PARP1 has long been associated exclusively with poly-ADPr, our antibodies revealed that mono-ADPr is in fact more abundant during the DNA damage response[21], illustrating the power of new tools to drive unexpected discoveries. This observation is explained by the rapid turnover of poly-ADPr, which constitutes an initial wave of PARP1 signalling, followed by a slower but more persistent accumulation of mono-ADPr[18]. This second signalling wave recruits RNF114, DTX3L and other proteins to DNA lesions[18]. RNF114, a ubiquitin E3 ligase that binds both mono-ADPr[18] and ubiquitin, serves as a reader of ADP-ribosyl-ubiquitylation (ADPrUb), an emerging composite PTM[31–34,39] we recently showed targets specific serine mono-ADPr marks on histones and PARP1 upon DNA damage[33].

Here, we extend our serine ADPr technology[20,21] to develop SpyTag-based antibodies for sensitive and specific poly-ADPr detection across multiple applications. Employing diverse serine ADPr substrates, we characterise and validate our poly-ADPr antibodies alongside affinity-matured mono-ADPr and H3S10/S28ADPr antibodies. Applying this toolset, we find that PARP1 hyperactivation resulting from XRCC1 deficiency increases not only poly-ADPr, as previously reported[10,11], but also mono-ADPr. In addition, our SpyTag ZUD reagent[33] reveals that XRCC1 loss elevates cellular levels of ADPrUb, enhancing RNF114 recruitment to sites of DNA damage.

## Results

### A 'mono-ADP-ribosylation blocking' strategy
Previously, we established a programmable PARP1/HPF1 reaction for the rapid and scalable generation of serine mono-ADPr peptides, which enabled the development, validation and characterisation of site-specific, mono-ADPr and pan-ADPr antibodies[18,20,21]. Here, we aimed to adapt our strategy to generate modular antibodies specific for poly-ADPr. Inspired by the possibility of extending the initial ADP-ribose attached to serine[40], we employed a simple protocol that does not require separation of different ADP-ribosylated species. Specifically, following the serine mono-ADPr reaction, we purified the mono-ADPr peptide from the reaction components, as in our original approach[21], and subsequently added PARP1, but not HPF1, to extend

the mono-ADP-ribose with additional ADP-ribose units (Fig. 1A and Supplementary Fig. 1A). While we generated almost purely mono-ADPr peptides[21] (Fig. 1B and Supplementary Fig. 1B, C), we were unable to further ADP-ribosylate more than half of the initial mono-ADPr peptide (Fig. 1C and Supplementary Fig. 1B, D). Therefore, we redirected our strategy from peptides to automodification of full-length PARP1, which results in its complete serine poly-ADPr[21]. However, this approach did not yield any antibodies, possibly due to interference from the highly abundant and long poly-ADP-ribose chains on PARP1 during its immobilisation for phage display. To circumvent these challenges, we returned to poly-ADPr peptides and drew inspiration from our successful guided selection strategy, in which the unmodified peptide steered the selection towards antibodies that recognise the ADP-ribose moiety rather than other regions of the target peptide[21]. The obvious extension of this strategy to poly-ADPr would involve using a poly-ADPr-modified peptide as the antigen and an unmodified counterpart peptide for blocking. However, this approach would require a pure poly-ADPr peptide, which we were unable to generate, so we reasoned that blocking with a mono-ADPr-modified peptide would prevent the selection of antibodies that recognise the abundant contaminating mono-ADPr species (Fig. 1A and Supplementary Fig. 1A). With this strategy, our ability to generate only partially poly-ADP-ribosylated peptides (Fig. 1C and Supplementary Fig. 1B) would be sufficient for the selection of antibodies specific for poly-ADPr.

Given the value of the SpyTag system in generating modular antibodies[18], we chose to implement it for our poly-ADPr antibodies. As illustrated in the scheme (Supplementary Fig. 1E, F), the monovalent antibody can be rapidly and reliably converted into a bivalent format using various catchers for coupling[38], such as an HRP-coupled antibody for immunoblotting or a synthetic IgG antibody for immunofluorescence[18,38].

### Diagnostic ions for identification of di-ADP-ribosylation sites
During our mass spectrometric analysis, we observed fragment ions that appeared exclusively in spectra of poly-ADPr peptides (Supplementary Fig. 1C, D). To test the hypothesis that these peaks represent specific diagnostic ions for poly-ADPr, we modified a second peptide and observed the same fragmentation pattern for its di- and tri-ADPr forms, but not for the mono-ADPr form (Fig. 2A). From the structure of di-ADP-ribose, we interpreted these ions as corresponding to singly charged AMP+phosphoribose, ADP+phosphoribose, ADPr+AMP and di-ADPr (Fig. 2A and Supplementary Fig. 2A, B). These additional ions are specific for poly-ADPr because their formation requires two covalently attached ADP-ribose moieties. In contrast, mono-ADPr diagnostic ions can originate either from the terminal ADP-ribose within a poly-ADPr chain or from mono-ADPr.

Given the importance of specific diagnostic ions in our recent approach to distinguish ADP-ribosyl-ubiquitylation from the co-occurrence of ADPr and ubiquitylation as separate modifications[33], we reasoned that poly-ADPr-specific diagnostic ions could likewise serve as an effective means to distinguish poly- from mono-ADPr. Therefore, building on our reanalysis strategy[41], we searched for di-ADP-ribosylated peptides in published EThcD data[42]. After manual validation of MS/MS spectra, we confidently identified di-ADPr on primary serine ADPr targets[14,15,42], including histones and PARP1 (Fig. 2B, C, Supplementary Fig. 2C and Supplementary Data 1).

Given the risk of site mislocalization in ADPr proteomics[14,43], we hypothesised that di-ADPr could contribute to this issue, as computational proteomics analyses typically consider only mono-ADPr. We reasoned that if search engines are not allowed to consider di-ADPr as a potential modification, they may be forced to assign two separate mono-ADPr modifications to peptides actually modified by a single di-ADPr. Consequently, a di-ADPr-modified residue could be misassigned as mono-ADP-ribosylated, with an adjacent residue incorrectly annotated as the second mono-ADPr site. To test whether interpreting di-

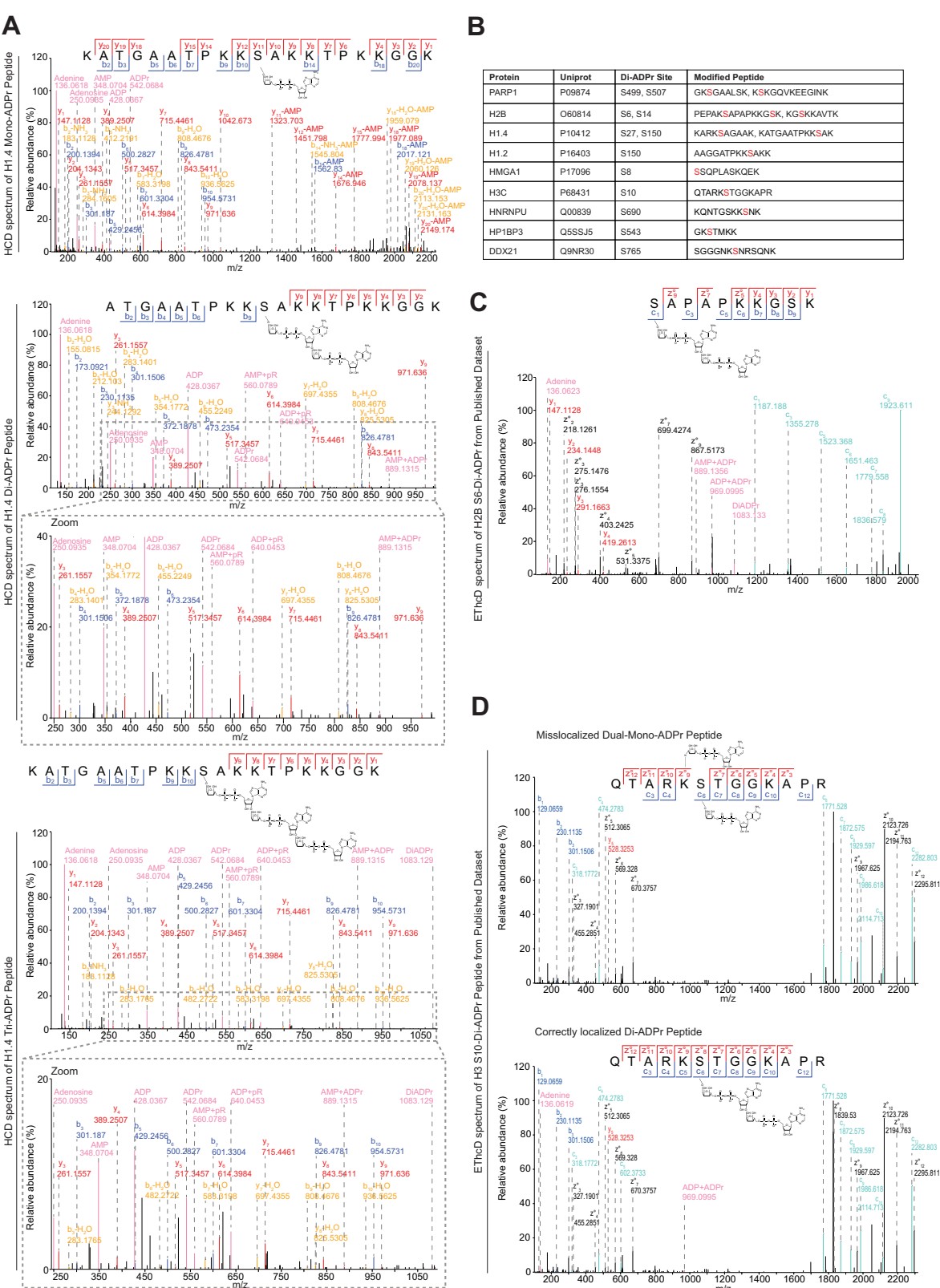

| Protein | Uniprot | Di-ADPr Site | Modified Peptide |
|---------|---------|--------------|------------------|
| PARP1 | P09874 | S499, S507 | GKSGAALSK, KSKGQVKEEGINK |
| H2B | O60814 | S6, S14 | PEPAKSAPAPKKGSK, KGSKKAVTK |
| H1.4 | P10412 | S27, S150 | KARKSAGAAK, KATGAATPKKSAK |
| H1.2 | P16403 | S150 | AAGGATPKKSAKK |
| HMGA1 | P17096 | S8 | SSQPLASKQEK |
| H3C | P68431 | S10 | QTARKSTGGKAPR |
| HNRNPU | Q00839 | S690 | KQNTGSKKSNK |
| HP1BP3 | Q5SSJ5 | S543 | GKSTMKK |
| DDX21 | Q9NR30 | S765 | SGGGNKSNRSQNK |

ADPr as two mono-ADPr modifications causes such mislocalization, we compared two reanalyses: one considering only mono-ADPr and another allowing both mono- and di-ADPr as variable modifications. We observed that restricting the search to mono-ADPr can result in misassignment, with ADPr incorrectly localised to a lysine instead of the adjacent serine modified by di-ADPr (Fig. 2D). In the search allowing for di-ADPr, the detection of poly-ADPr-specific diagnostic ions, along with additional fragment ions between the lysine and serine – unannotated after the mono-ADPr-only search – demonstrates that only the serine residue is modified by di-ADPr.

Overall, the diagnostic ions identified from peptide antigens designed for poly-ADPr antibody generation enabled mapping of di-ADPr sites in proteomics analyses, correcting mislocalizations inherent to standard workflows that consider only mono-ADPr.

**Fig. 2 | Validation of poly-ADPr diagnostic ions and identification of di-ADPr sites on various proteins including histones and PARP1. A** Spectra of an in vitro modified H1.4 peptide. The peptide was first mono-ADP-ribosylated and subsequently poly-ADP-ribosylated. Both peptides were analysed by mass spectrometry. Spectra of mono-, di- and tri-ADP-ribosylated H1.4 are shown and the corresponding diagnostic ions of mono- or poly-ADPr are labelled in pink. $N = 1$. **B** Table of identified di-ADPr sites in the analysed public data set. Multiple sites on different proteins were detected. The corresponding spectra were first filtered to have a Score > 40, Delta Score > 20 and a localisation probability > 0.8 followed by inspection of annotated spectra in MaxQuant, and only included if the modification site was covered by sequence ions. **C** Annotated EThcD spectrum of H2B modified by di-ADPr on serine 6. The poly-ADPr diagnostic ions were annotated manually if present. $N = 1$. **D** Di-ADPr can be misslocalised as two different mono-ADPr sites. The same spectrum of two separate MaxQuant analysis is shown. The upper spectrum is the result of an analysis in which di-ADPr was not allowed as a variable modification, leading to a reasonably good spectrum with two mono-ADPr sites annotated on lysine 9 and serine 10 of H3, however the modification site is not covered by sequence ions. The lower spectrum is the result of an analysis in which di-ADPr and mono-ADPr were allowed as variable modifications simultaneously. This increased the sequence coverage of the peptide, strikingly, the missing lysine and serine ions are detected if di-ADPr is allowed. The mass of di-ADPr is clearly localised to serine 10, demonstrating, that the peptide fragmented in this spectrum is modified with a di-ADPr on serine 10 instead of two single mono-ADPr units. $N = 1$.

## Modular Antibodies for versatile and sensitive detection of Poly-ADPr

To generate modular antibodies specific for poly-ADPr, we employed a mixture of mono- and poly-ADP-ribosylated histone H3S10 peptides (Fig. 1C and Supplementary Fig. 1B). Building on our previous phage display approach[21], we introduced a key modification: guided selection toward poly-ADPr through our mono-ADPr blocking strategy (Fig. 1A and Supplementary Fig. 1A).

We first assessed the specificity of two selected poly-ADPr antibodies (AbD64138, AbD64235) by taking advantage of a feature of serine ADPr: the inability of PARG to cleave the bond between serine and the initial ADP-ribose[21], despite its ability to hydrolyse tyrosine and aspartate/glutamate ADPr[17,44]. Specifically, we automodified PARP1 in the presence of HPF1, resulting in nearly complete poly-ADPr of PARP1. Addition of PARG to the reaction completely reduced poly-ADP-ribosylated PARP1 to mono-ADP-ribosylated PARP1[21], resulting in mono- and poly-ADPr standards for antibody validation. While our mono-ADPr antibody[18] recognised only mono-ADP-ribosylated PARP1, our poly-ADPr antibodies showed no reactivity toward mono-ADPr but produced a strong signal for poly-ADP-ribosylated PARP1 (Fig. 3A and Supplementary Fig. 3A). As expected, the 10H antibody and the WWE-based reagent[45] proved specific for poly-ADPr, although 10H detected only long ADPr polymers (Fig. 3A). Similar to the WWE-based reagent, our antibodies detected the full spectrum of poly-ADPr chain lengths in immunoblotting. To ensure a fair comparison of signal intensities, we tested the reagents at two different dilutions, starting with those recommended for each tool. However, using our antibodies at the same concentration (1 μg/ml) immediately caused signal bleaching. Even after reducing the antibody concentration tenfold, the sensitivity of our antibodies remained so high that oversaturation occurred after just one second of exposure. To enable a more accurate comparison without signal saturation, we performed a dot blot using a 1:2 dilution series of poly-ADP-ribosylated PARP1 (Fig. 3B and Supplementary Fig. 3B, C). Antibodies were tested at either the recommended concentrations or at a common dilution of 0.1 μg/ml. In the HRP-SpyTag format, our antibodies produced significantly stronger signal intensities than both the WWE-based reagent and the CST pan-ADPr antibody, while the 10H antibody failed to generate any detectable signal at this dilution.

Our antibodies are not restricted to serine-ADPr but also detect poly-ADPr on aspartate and glutamate, indicating that they recognise poly-ADPr irrespective of the underlying amino acid (Fig. 3C and Supplementary Fig. 3D). By establishing an optimised electrophoretic separation method for resolving oligo-ADP-ribosylated peptides, we found that the antibodies recognise poly-ADPr chains as short as di-ADPr (Supplementary Fig. 3E). Moreover, they efficiently detect free poly-ADP-ribose (Supplementary Fig. 3F). Together, these results illustrate the broad substrate recognition of our poly-ADPr antibodies.

Next, we assessed the performance of these tools in cells treated with $H_2O_2$ for ten minutes, with or without the PARP inhibitor Olaparib (Fig. 3D and Supplementary Fig. 3G). All reagents were used at their optimal dilutions. No nonspecific binding was detected, as evidenced by the absence of signal in Olaparib-treated samples (lanes 2 and 4 of each blot). Our antibodies exhibited high sensitivity, even when compared to commercially available reagents used at tenfold higher concentrations. Importantly, even without PARG inhibition previously required for detection[46], this exceptional sensitivity enables detection of poly-ADPr in unperturbed cells (Fig. 3E and Supplementary Fig. 3H), where its generation has been linked to unligated Okazaki fragments[46]. This sensitivity is particularly critical when basal ADPr levels are low, such as in RPE1 cells grown under hypoxia. Even under these challenging conditions, our antibodies detected PARP1-dependent poly-ADPr, whereas commercial reagents failed to produce a detectable signal (Fig. 3E and Supplementary Fig. 3H).

To further assess specificity, we performed immunofluorescence microscopy using the synthetic IgG format. We tailored dilutions and microscope settings to maximise the performance of each reagent. While signal intensities are not directly comparable, our antibodies provided a better signal-to-noise ratio in untreated versus $H_2O_2$-treated cells (Fig. 3F and Supplementary Fig. 4A) and performed best with methanol fixation (Supplementary Fig. 4B). Given the strong influence of antibody dilution on background signal, we tested multiple dilutions, all of which maintained poly-ADPr specificity with minimal nonspecific binding (Supplementary Fig. 4C). Next, inspired by the simultaneous detection of mono- and poly-ADPr using ADP-ribose binding reagents[47], we leveraged the format switching enabled by SpyTag technology to detect these ADPr forms concurrently in cells. Specifically, we coupled the poly-ADPr antibody with a mouse Fc SpyCatcher and the mono-ADPr antibody with a rabbit Fc SpyCatcher. To prevent channel bleed-through, we used two fluorophores with distinct emission peaks and employed line sequencing during imaging. The results demonstrate that, following a ten-minute $H_2O_2$ treatment, both mono- and poly-ADPr can be detected simultaneously in the same cells (Supplementary Fig. 4D). Poly-ADPr staining appeared as nuclear foci, suggesting the localised activation of PARP enzymes at specific chromatin sites, where polymeric ADP-ribose accumulates. In contrast, mono-ADPr exhibited a more diffuse nuclear distribution, consistent with its broader roles across chromatin.

## Affinity-matured mono-ADPr and H3S10/S28ADPr antibodies

Whereas our poly-ADPr antibodies are both sensitive and specific as original clones (Fig. 3), our other antibodies have shown specificity but limited sensitivity[21]. Although the original clones of these antibodies have advanced studies across multiple laboratories[22,23,28,30,35,48], the affinity maturation of AbD33204[21] into AbD43647[18] demonstrates how increased sensitivity can broaden their applicability[17,23,24,27,36]. Therefore, to increase the sensitivity of our site-specific H3S10/S28ADPr antibody AbD33644 and our mono-ADPr antibody AbD33205[21], we applied the affinity maturation strategy (Fig. 4A) that yielded AbD43647[18]. While both AbD33205 and AbD33204/AbD43647 are specific for mono-ADPr, they exhibit differences in substrate preference[18,21], making them complementary and valuable for different applications[17,22,23,27,28,30,35,36,48]. For example, only AbD33204/AbD43647, but not AbD33205, recognise free ADP-ribose[18,21].

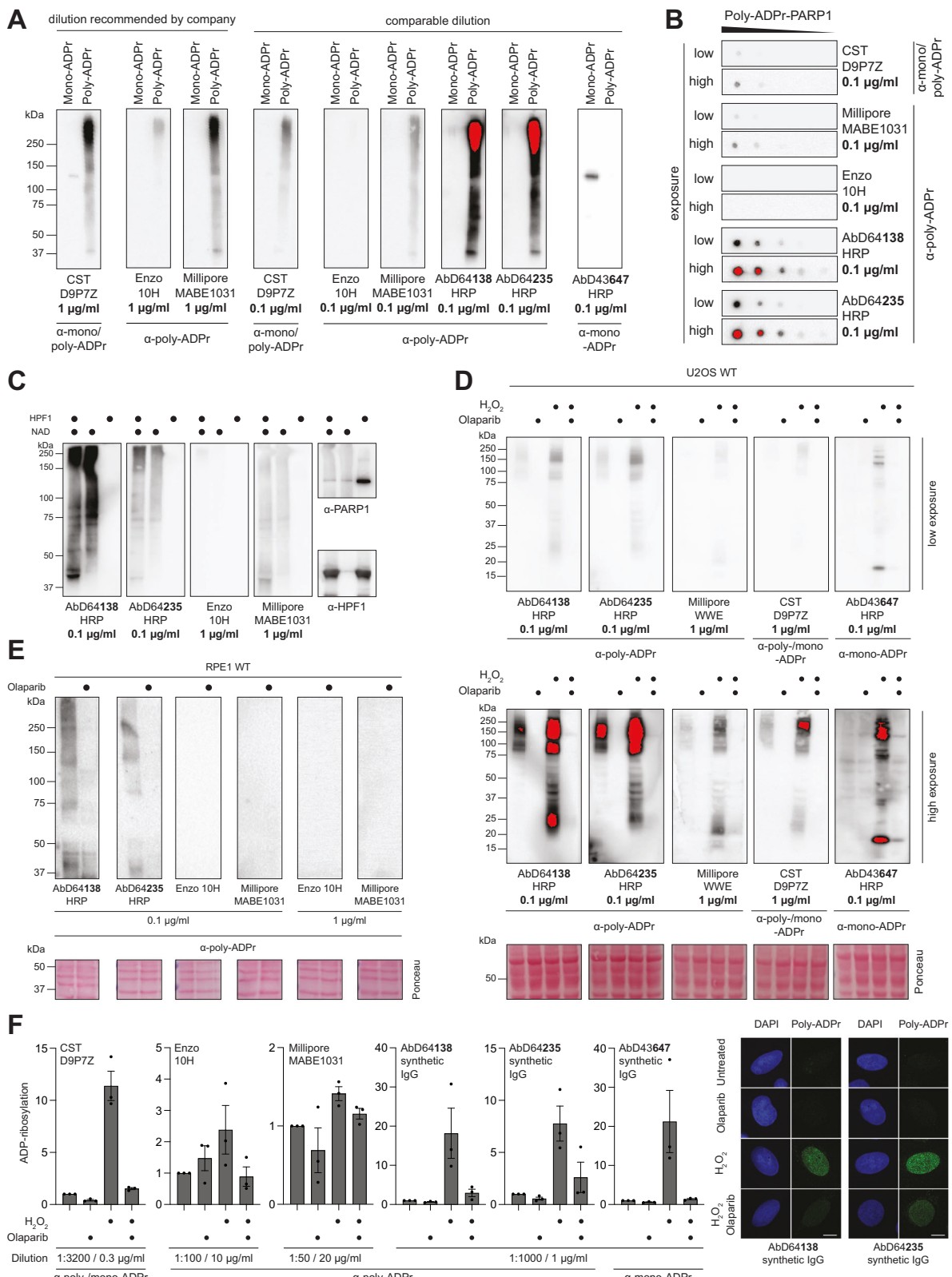

We performed ELISA analyses on the selected antibodies – the new mono-ADPr clone AbD41122 and the new H3S10/S28 clones AbD55568 and AbD55558 – using PARP1S507, H3S10, H3S28, and H1.4S149 mono-ADPr peptides[21], unmodified H3(1-21) peptide and free mono-ADP-ribose (Fig. 4B). As with our previously developed antibodies[18,21], none of the selected clones bound the unmodified peptide. Like its parental clone[21], AbD41122 did not recognise free

mono-ADP-ribose and showed distinct context dependencies (Fig. 4B). AbD41122 and AbD43647 differ substantially in substrate preference, and no single clone is sufficient to capture the full diversity of mono-ADPr (Supplementary Fig. 5A). Moreover, AbD41122 robustly detects starvation-induced mono-ADPr on RNA[49], while AbD43647 does so only weakly and with nonspecific background (Fig. 4C and Supplementary Fig. 5C), making AbD41122 the preferred reagent for specific

**Fig. 3 | Modular antibodies for versatile, specific and sensitive detection of Poly-ADPr. A** Immunoblot analysis of in vitro ADP-ribosylated PARP1 with (mono-ADPr) or without (poly-ADPr) PARG using Poly-/ Pan-/ Mono-ADPr antibodies. Detection was done in parallel resulting in comparable signal intensities. Red colour represents saturated signal. $N = 3$. **B** Dot-blot analysis of in vitro auto-poly-ADPr of PARP1 in 1:2 dilution series using Poly-/ Pan-ADPr antibodies. Detection was done in parallel resulting in comparable signal intensities. Red colour represents saturated signal. $N = 3$. **C** Immunoblot analysis of in vitro ADP-ribosylated PARP1 with (Ser-ADPr) or without (Asp/Glut-ADPr) HPF1 using Poly-ADPr antibodies. Detection was done in parallel resulting in comparable signal intensities. $N = 3$. **D** Immunoblot analysis of SDS cell extracts from 2 mM H2O2-treated wild-type (WT) U2OS cells with and without 1 μM Olaparib treatment with the indicated antibodies. Detection was done in parallel resulting in comparable signal intensities. Red colour represents saturated signal. $N = 3$. **E** Immunoblot analysis of SDS cell extracts from untreated wild-type (WT) hTERT RPE1 cells with and without 1 μM Olaparib treatment for 1 h with the indicated antibodies. Detection was done in parallel, resulting in comparable signal intensities. $N = 3$. **F** Immunofluorescent staining of mono-/ poly-/ pan-ADPr in 2 mM H2O2-treated WT U2OS cells with and without 1 μM Olaparib treatment using the indicated antibodies. Detection was done on individual basis, signal intensities between different antibodies are not comparable. Signals are normalised to WT untreated conditions. Error bars represent SEM. Combined analysis of 3 Biological Replicates. Representative Images from 1 Biological Replicate. Scale bar, 10 μM. Source data are provided as a Source Data file.

detection of mono-ADPr on RNA. The two H3S10/S28 clones retained the specificity of their parental clone AbD33644[21], primarily recognising the H3S10ADPr site and, to a lesser extent, the less abundant H3S28ADPr site (Fig. 4B and Supplementary Fig. 5B), consistent with the sequence similarities surrounding these two sites.

Next, we used AbD41122 for immunoblotting and immunofluorescence of cells treated with DNA-damaging agents and the PARP inhibitor Olaparib. This new antibody was more sensitive than its parental clone AbD33205[21] and comparable to AbD43647[18], although it recognised a different pattern (Fig. 4D, E and Supplementary Fig. 5D, E). The new H3S10/S28ADPr clones were more sensitive than their parental clone but exhibited similar background signal, which was largely eliminated by sulfuric acid histone extraction (Fig. 4F and Supplementary Fig. 5F–H).

Next, we treated cells with a PARG inhibitor, which massively increases cellular poly-ADPr levels, as previously used to demonstrate the mono-ADPr specificity of antibodies[18]. While PARG inhibition led to a strong increase in signal detected by both pan- and poly-ADPr antibodies, the AbD41122 signal did not increase, confirming its mono-ADPr specificity (Fig. 4G and Supplementary Fig. 5I). The distinct staining pattern in response to PARG inhibition compared to AbD43647 further illustrates the complementarity of these antibodies. For the site-specific antibodies, we observed a smear above the main H3 band, suggesting that these clones recognise both mono- and poly-ADPr on H3S10/S28 (Fig. 4G).

The ability to convert these modular antibodies into bivalent biotin-coupled reagents via SpyTag/SpyCatcher ligation – offering improved pull-down efficiency compared to IgG (Supplementary Fig. 5J) – makes AbD41122 suitable for mono-ADPr enrichment (Supplementary Fig. 5K). Combining the biotin SpyTag format for pulldown with the HRP SpyTag format for immunoblotting can enhance signal detection and avoid antibody leakage[18,33].

## XRCC1 deficiency elevates the mono-ADPr wave of PARP1 signalling

Having developed modular antibodies for the specific and sensitive detection of mono-ADPr and poly-ADPr in cellular contexts (Figs. 3, 4 and Supplementary Fig. 5L), we proceeded to explore the different forms of ADPr generated by PARP1 following its hyperactivation induced by XRCC1 deficiency[10,11]. Our discovery of mono-ADPr as a second wave of PARP1 signalling[18], together with our tools, allows us to move technically and conceptually beyond the detection of pan-ADPr signals, which, following DNA damage, have often been assumed to represent poly-ADPr, including in the context of XRCC1 deficiency[11]. Given that ARH3 deficiency – which, like XRCC1 mutation, causes neurodegeneration – massively increases cellular mono-ADPr levels[21,30,50], we were intrigued to test whether XRCC1 deficiency affects mono-ADPr.

Consistent with a report that XRCC1 prevents PARP1 trapping[10], we and others observed an enhanced accumulation of endogenous PARP1 at DNA lesions in XRCC1 KO cells compared to WT (Supplementary Fig. 6A)[51]. As the effect of XRCC1 deficiency on ADPr was

previously studied after recovery from H2O2-induced DNA damage[11], we first examined mono- and poly-ADPr under these conditions. As anticipated[11], poly-ADPr levels, particularly on histones, increased in DNA-damaged XRCC1-deficient cells compared to WT (Fig. 5A and Supplementary Fig. 6B). In addition, we observed a marked increase in mono-ADPr, with the two antibodies detecting distinct bands, as expected from their complementary specificities (Fig. 4). We observed higher mono- and poly-ADPr levels in WT compared to XRCC1 KO cells after recovery from methyl methanesulfonate (MMS) treatment (Supplementary Fig. 6C). This late decrease in PARP1 activity in XRCC1-deficient cells is consistent with the model in which early PARP1 hyperactivation consumes NAD+, thereby limiting auto-modification and polymer extension at later time points[10].

Next, as previously done for ARH3 deficiency[21], we investigated ADPr during the active phase of DNA damage – a condition under which a previous study using a pan-ADPr antibody did not detect differences between XRCC1 KO and WT cells[11]. Building on our prior evidence highlighting the importance of timing in ADPr pattern dynamics following DNA damage[17,18,21], we applied continuous H2O2 and MMS exposure and observed that the absence of XRCC1 elevates not only poly- but also mono-ADPr even during ongoing DNA damage (Fig. 5B and Supplementary Fig. 6D, E). However, after 90 min of MMS-induced DNA damage, WT cells showed higher levels of poly-ADPr and core histone mono-ADPr than XRCC1 KO cells (Supplementary Fig. 6E), in line with reduced PARP1 activity at later stages of damage in XRCC1-deficient cells[10].

Both forms of ADPr are generated by PARP1, as their signals were largely abolished by PARP1 inhibition using either Olaparib or the PARP1-specific inhibitor AZD9574 (Fig. 5C), yet persisted after boiling (Supplementary Fig. 6F), suggesting modification on serine residues[17]. We were intrigued by a ~ 25–30 kDa band that was strongly increased in XRCC1 KO cells following both H2O2 and MMS treatments, and was more prominently detected by AbD41122 than by AbD43647 (Fig. 5A, B and Supplementary Fig. 6B, E). This band is interesting since mono-ADPr on this target is not a remnant of poly-ADPr, given that it is unaffected by PARG inhibition[18]. Through perchloric acid enrichment, which solubilises histone H1 while leaving most other proteins insoluble (Supplementary Fig. 6G), we identified this mono-ADPr–exclusive target as histone H1 (Fig. 5D). This identification was further confirmed by mutating the main ADPr sites on H1[15,42] (Fig. 5E). The sustained mono-ADPr signal on H1 in XRCC1-deficient cells is intriguing, as studies indicate that mono-ADPr weakens H1–DNA interactions and can promote its displacement from chromatin during DNA damage[52], a process that may contribute to chromatin relaxation at damage sites. Prolonged H1 mono-ADPr may therefore reflect extended chromatin remodelling resulting from persistent repair intermediates.

Next, we investigated the behaviour of mono- and poly-ADPr in response to XRCC1 deficiency and various genotoxic treatments. Exploring ADPr dynamics is essential to accurately define how distinct forms of ADPr respond to different treatments and genetic perturbations, as these can influence not only the extent but also the timing of

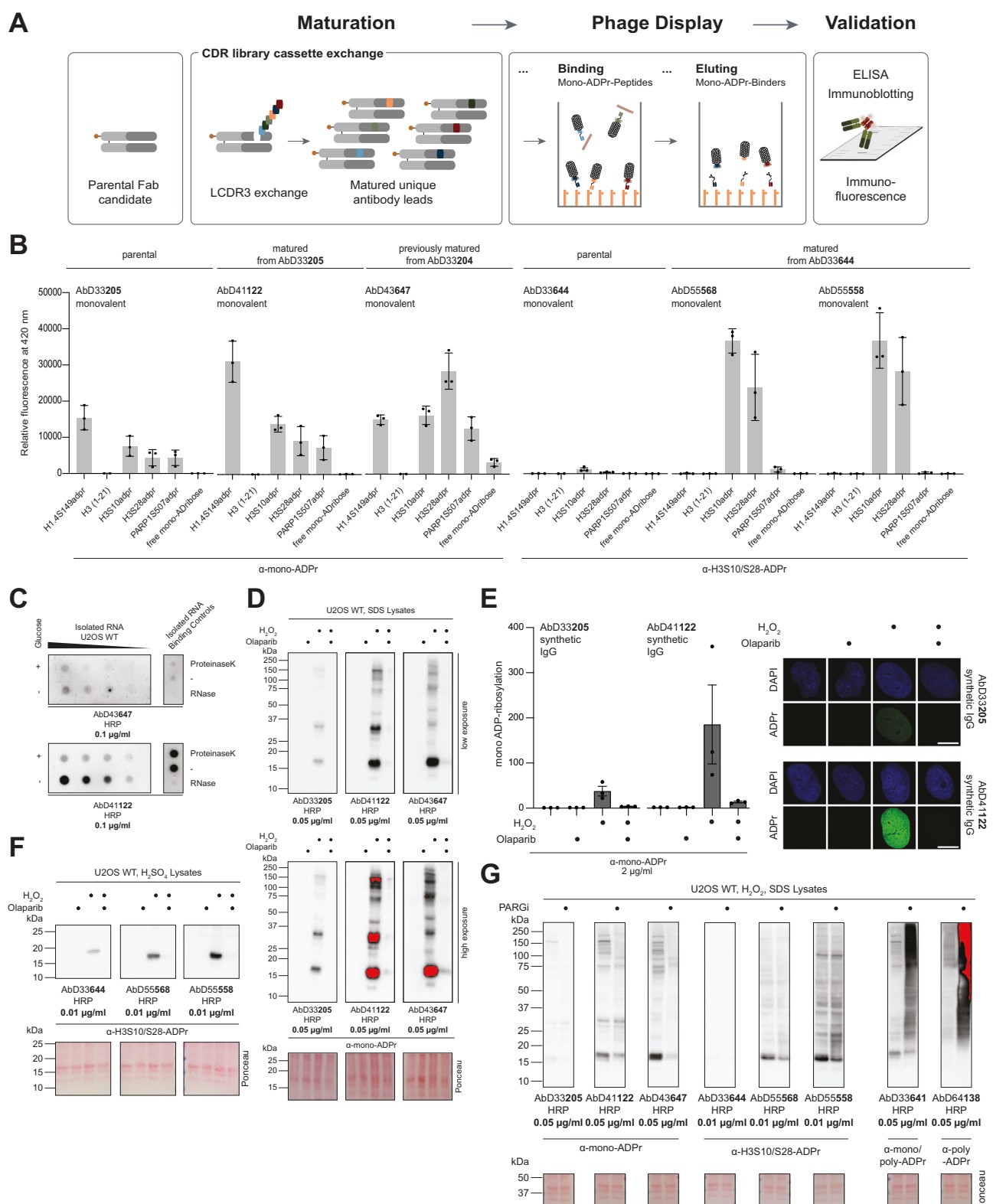

ADPr induction, as we have shown previously[17,18,21]. We first applied our live-cell imaging approach to monitor the recruitment of mono- and poly-ADPr probes to sites of DNA damage[18]. XRCC1 deficiency led to a substantial rise in poly-ADPr and mono-ADPr, especially at later time points (Fig. 6A), consistent with our immunoblotting results (Fig. 5).

To further dissect the dynamics of poly- and mono-ADPr, we treated XRCC1 KO and WT cells with distinct types of DNA damage – $H_2O_2$, MMS and camptothecin (CPT) – across varying durations.

Intriguingly, XRCC1 deficiency alters ADPr dynamics in a damage type–specific manner (Fig. 6B, C). CPT primarily induced poly-ADPr, with mono-ADPr detectable only in XRCC1-deficient cells. $H_2O_2$ led to an overall elevation of both mono- and poly-ADPr, whereas MMS induced a much more rapid ADPr response in XRCC1-deficient cells compared to WT (Fig. 6B, C and Supplementary Fig. 7A, B). We observed a similar ADPr pattern in XRCC1-deficient patient fibroblasts (Supplementary Fig. 7C). Overall, mono-ADPr appeared more stable

**Fig. 4 | Affinity-matured H3S10/S28 and mono-ADPr antibodies. A** Schematic illustration of the affinity maturation process. Illustrations generated in Adobe Illustrator. Schematics adapted from Dauben et al.[20]. **B** ELISA analysis of antibody specificities using the indicated antibodies (2 µg/mL) and biotinylated peptides (61 nM). Bars represent the arithmetic mean of 3 independent experiments. Error bars represent SD. **C** Left: Dot Blot analysis of isolated RNA from U2OS cells before and after glucose starvation. Right: Enzymatic digestion control of isolated RNA. Detection was done in parallel, resulting in comparable signal intensities. $N = 3$. **D** Immunoblot analysis of SDS cell extracts from 2 mM H2O2-treated wild-type (WT) U2OS cells with and without 1 µM Olaparib treatment with the indicated antibodies. Detection was done in parallel, resulting in comparable signal intensities. Red colour represents a saturated signal. $N = 3$. **E** Immunofluorescent staining of mono-ADPr in 2 mM H2O2-treated WT U2OS cells with and without 1 uM

Olaparib treatment using the indicated antibodies. Detection was done in parallel, resulting in comparable signal intensities. Signals are normalised to WT untreated conditions. Error bars represent SEM. Combined Analysis of 3 Biological Replicates. Representative Images from 1 Biological Replicate. Scale bar, 10 µM. **F** Immunoblot analysis of H2SO4 cell extracts from 2 mM H2O2-treated wild-type (WT) U2OS cells with and without 1 µM Olaparib treatment with H3S10/S28-ADPr site-specific antibodies. Detection was done in parallel, resulting in comparable signal intensities. $N = 3$. **G** Immunoblot analysis of SDS cell extracts from 2 mM H2O2-treated wild-type (WT) U2OS cells with and without 1 µM PARG inhibition treatment with the indicated antibodies. Detection was done in parallel, resulting in comparable signal intensities. Red colour represents a saturated signal. $N = 3$. Source data are provided as a Source Data file.

and homogeneously distributed among the cell population, while poly-ADPr remained transient and spatially heterogeneous – particularly after H2O2 treatment (Fig. 6B). This more heterogeneous distribution of poly-ADPr likely reflects its transient nature, whereby rapid turnover renders its detection highly sensitive to subtle temporal or cell-to-cell differences in PARP1 activity. Immunoblotting further revealed treatment-specific temporal dynamics: H2O2 induced distinct sequential waves of mono- and poly-ADPr; with MMS, the waves were temporally closer; and with CPT, they largely overlapped, resulting in low-levels of mono-ADPr detectable only under conditions of PARP1 hyperactivation caused by XRCC1 loss (Fig. 6C, D). To contextualise XRCC1 deficiency, we compared it with ARH3 loss, as both are linked to neurodegeneration and share elevated mono-ADPr (Figs. 5, 6)[21,30,50]. While both conditions lead to increased mono-ADPr, ARH3 deficiency affects exclusively mono-ADPr. In contrast, XRCC1 loss increases both mono- and poly-ADPr, displaying a distinct pattern compared to ARH3-deficient cells (Supplementary Fig. 7D), consistent with elevated mono-ADPr arising from altered PARP1 activation rather than impaired mono-ADPr removal.

Collectively, these findings demonstrate the profound impact of XRCC1 deficiency not only on the magnitude but also on the timing of ADPr (Fig. 6D). Importantly, this includes mono-ADPr – the long-overlooked second wave of PARP1 activity[18] – which we can now directly visualise (Fig. 4). Crucially, the increase in both mono- and poly-ADPr caused by XRCC1 loss allowed us to apply our tools to uncover how different genotoxic agents distinctly affect these two signalling waves.

### Altered mono-ADPr dynamics in XRCC1-deficient cells involve PARG-dependent turnover

Having observed that mono-ADPr levels increase in the absence of XRCC1 (Figs. 5, 6), we next investigated whether this signal arises from altered PARG-dependent turnover of poly-ADPr. To address this, we examined the effect of PARG inhibition (PARGi) on the accumulation of serine mono-ADPr in XRCC1 KO cells. Although PARG can remove mono-ADPr from tyrosine and aspartate/glutamate[17,44], it cannot hydrolyse serine-linked mono-ADPr[21]. PARG inhibition, therefore, provides a means to assess how much serine mono-ADPr arises from poly-ADPr degradation. Consistent with our previous studies[18,21], when PARG inhibition is applied throughout genotoxic treatment, mono-ADPr is largely PARG-dependent, as evidenced by the strong reduction in mono-ADPr signal (Fig. 7A and Supplementary Fig. 7E). Notably, the extent of this reduction is similar in WT and XRCC1 KO cells, consistent with elevated mono-ADPr levels in XRCC1 KO cells arising from increased levels of the PARG substrate poly-ADPr rather than from increased PARG activity.

Next, we extended the late-stage PARG inhibition approach of Demin et al.[10] by applying short PARGi pulses (5, 10, or 15 min before harvest) during continuous 45 min H2O2 treatment, the condition showing the largest XRCC1-dependent increase in mono-ADPr relative to WT cells. This strategy provides a direct readout of acute PARG-

dependent conversion of poly-ADPr to mono-ADPr, allowing us to determine whether persistent mono-ADPr observed at late time points in XRCC1 KO cells is generated by rapid hydrolysis of pre-formed poly-ADPr. Consistent with low poly-ADPr levels at this late stage of ADPr signalling, short pulses of PARG inhibition had only a mild effect on mono-ADPr (Fig. 7B).

Together, our results provide no evidence that XRCC1 deficiency measurably alters PARG activity. Instead, they indicate that mono-ADPr accumulation in XRCC1-deficient cells reflects a time-dependent contribution of PARG-mediated turnover, with strong dependence on PARG activity at early stages revealed by full PARG inhibition (Fig. 7A), but only limited sensitivity to late-stage PARG inhibition (Fig. 7B).

### XRCC1 deficiency increases RNF114 accumulation at sites of DNA damage and elevates cellular levels of DNA damage-induced ADP-ribosyl-ubiquitylation

Our finding that XRCC1 deficiency leads to elevated levels of DNA damage-induced mono-ADPr prompted us to investigate the downstream consequences of this increase. Specifically, we tested the hypothesis that XRCC1 loss enhances recruitment of the ubiquitin E3 ligase RNF114 to sites of DNA damage, based on our finding that RNF114 recruitment increases when mono-ADPr levels are elevated[18]. Using live-cell imaging to track GFP-tagged RNF114[18], we observed increased accumulation at DNA lesions in XRCC1-deficient cells compared to WT cells (Fig. 8A). Given that RNF114 binds not only mono-ADPr[18] but also ubiquitin, functioning as a reader of ADPrUb[31–34], we reasoned that XRCC1 deficiency might lead to increased cellular levels of this composite modification. In this context, we recently demonstrated that DNA damage-induced serine mono-ADPr marks on histones and PARP1 are further modified by this unconventional ester-linked ubiquitylation[33]. We were particularly encouraged by our observation that ADPrUb on serine is increased in cells lacking ARH3[33], a condition that leads to a marked accumulation of mono-ADPr[21,30,50] (Supplementary Fig. 7D).

To detect cellular ADPrUb, we employed our recently developed method combining biotin-coupled SpyTag-ZUD pulldown followed by immunoblotting with mono-ADPr antibodies and HRP-coupled Spy-Tag-ZUD[33]. While our mono-ADPr antibodies have been widely used to explore this composite modification in cells[31–33,36] – requiring minimal material for sensitive detection – ZUD, the first and only reagent specific for ADPrUb, remains considerably more challenging to use, requiring larger sample amounts and producing higher background[33]. Detection specificity was ensured using hydroxylamine, which cleaves the ester bond to release ubiquitin from mono-ADPr[17,33]. ZUD and mono-ADPr antibodies revealed a significant increase in DNA damage-induced ADPrUb in XRCC1 KO cells compared to WT cells (Fig. 8B), consistent with the enhanced recruitment of the ADPrUb reader RNF114 to DNA lesions (Fig. 8A). This highlights the impact of XRCC1 on ADPrUb signalling during the DNA damage response and could imply a link between XRCC1 and the various effects suggested for

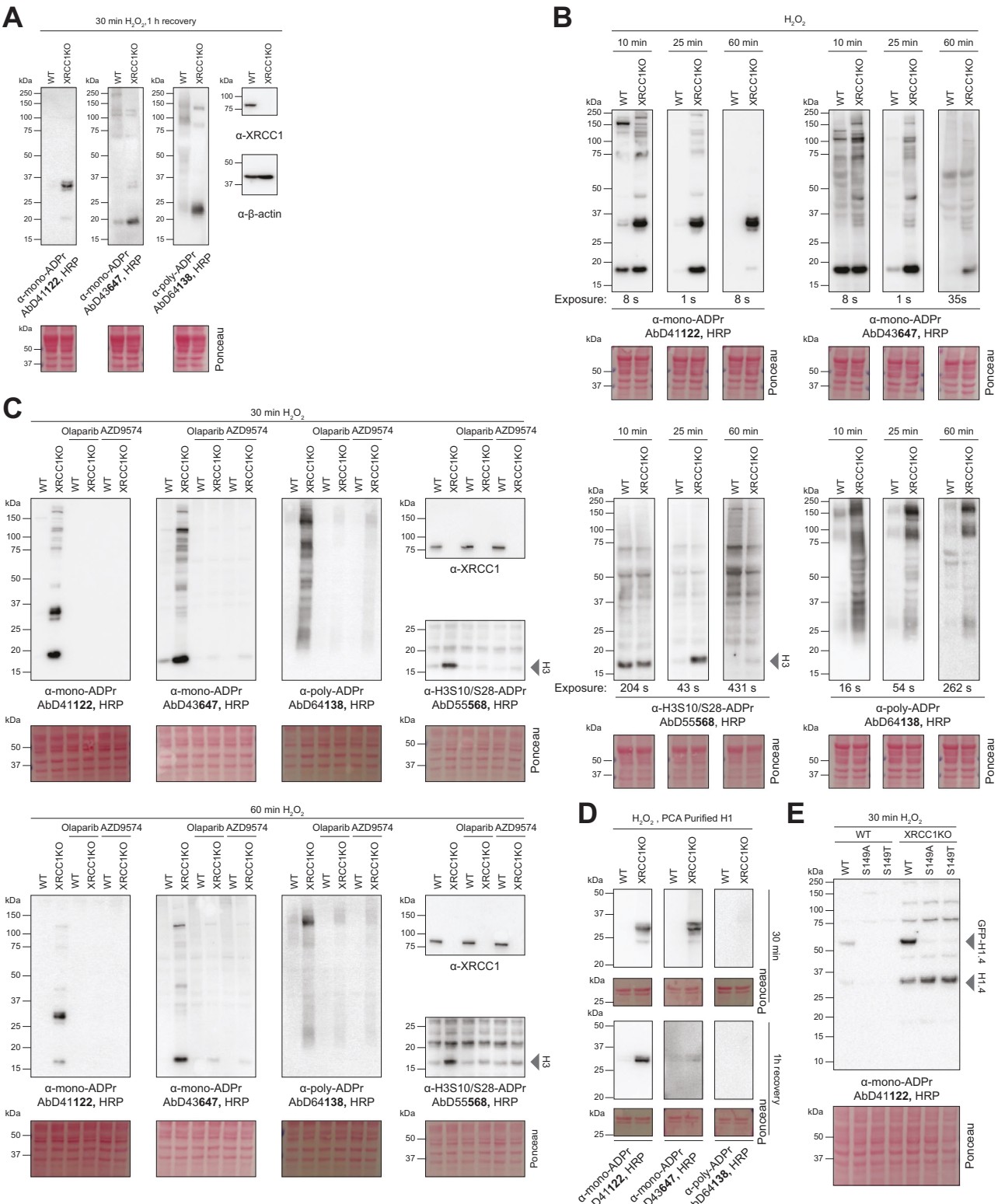

**Fig. 5 | XRCC1 deficiency elevates mono-ADPr wave of PARP1 signalling.**
**A** Immunoblot analysis of SDS cell extracts from 30 min 150 µM H2O2-treated followed by 1 h drug-free medium recovered wild-type (WT) and XRCC1KO hTERT RPE1 cells with the indicated antibodies. $N = 4$. **B** Immunoblot analysis of SDS cell extracts from continuously 150 µM H2O2-treated wild-type (WT) and XRCC1KO hTERT RPE1 cells for the indicated time points with the indicated antibodies. Time points are shown in separate panels because different exposure times were required to accurately capture signal intensities, as indicated. $N = 5$. **C** Immunoblot

analysis of SDS cell extracts from 150 µM H2O2-treated wild-type (WT) and XRCC1KO hTERT RPE1 cells with and without PARP1 inhibition (1 µM Olaparib or 10 nM AZD9574) for the indicated time points with the indicated antibodies. $N = 3$. **D** Immunoblot analysis of isolated Histone H1 using three step perchloric acid (PCA) lysis of WT and XRCC1KO hTERT RPE1 cells after 30 min 150 µM H2O2-treated followed or not by 1 h drug free medium recovery. $N = 3$. **E** Immunoblot analysis of WT and XRCC1KO hTERT RPE1 cells after 30 min 150 µM H2O2-treated with H1.4-GFP overexpression. $N = 3$. Source data are provided as a Source Data file.

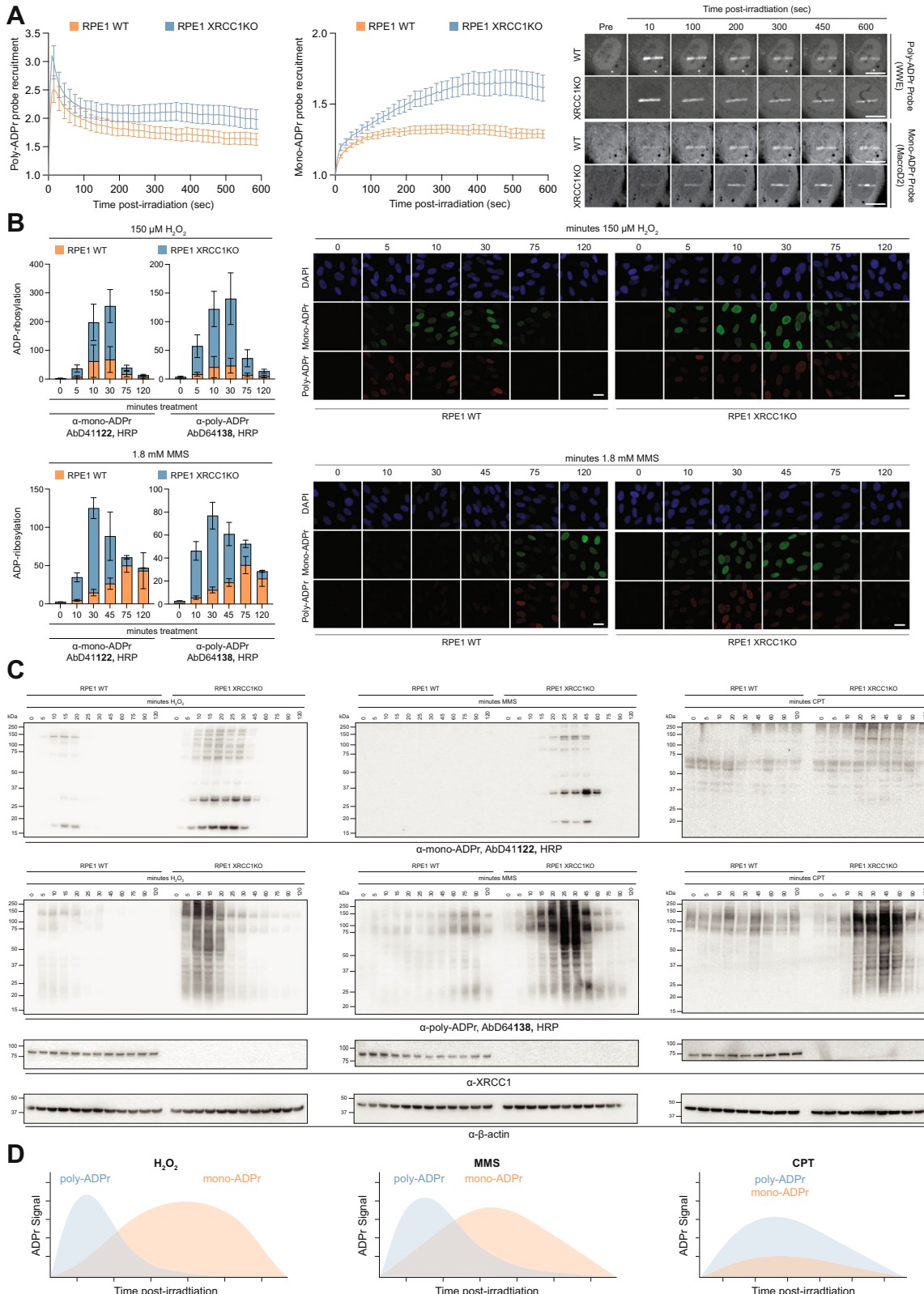

**Fig. 6 | Effect of XRCC1 deficiency on cellular mono- and poly-ADPr dynamics upon DNA damage. A** Real-time live-cell detection of Poly- (left) and Mono-ADPr (middle) in hTERT RPE1 WT or XRCC1KO cells. Representative recruitment kinetics (left) and confocal images (right). Error bars represent SEM. Scale bar, 10 μM. $N = 3$. **B** Immunofluorescent co-staining of mono- and poly-ADPr in hTERT RPE1 WT or XRCC1KO cells followed by 150 μM H2O2 or 1.8 mM MMS treatment for the indicated time points. Left, quantitative analysis with error bars representing SEM. Signals are normalised to WT untreated conditions. Error bars represent SEM.

Combined Analysis of 3 Biological Replicates (left). Representative Images from 1 Biological Replicate (right). Scale bar, 10 μM. **C** Immunoblot analysis of SDS cell extracts from continuously 150 μM H2O2, 1.8 mM MMS or 15 μM CPT-treated wild-type (WT) and XRCC1KO hTERT RPE1 cells for the indicated time points with the indicated antibodies. $N = 3$. **D** Schematic illustration of XRCC1 deficiency-driven reshaping of mono- and poly-ADPr waves under different DNA damage-inducing reagents. Source data are provided as a Source Data file.

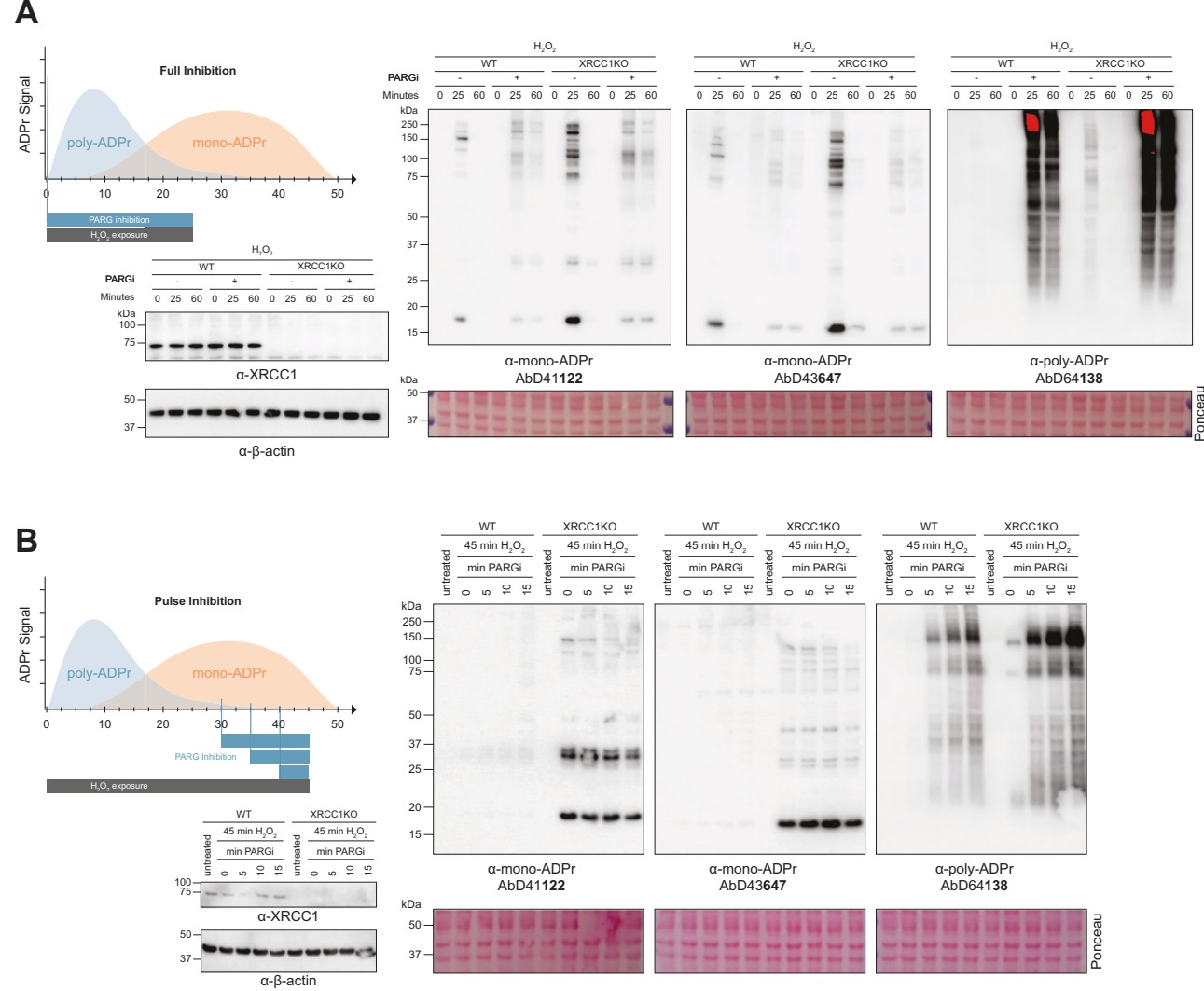

**Fig. 7 | Altered mono-ADPr dynamics in XRCC1-deficient cells involve PARG-dependent turnover. A** Illustration showing the experimental set-up for full PARG inhibition (left top). Immunoblot analysis of SDS cell extracts from 150 μM H2O2-treated wild-type (WT) and XRCC1KO hTERT RPE1 cells with and without 1 μM PARG inhibition for the indicated time points with the indicated antibodies (right). $N = 3$.

**B** Illustration showing experimental set-up for pulse PARG inhibition (left top). Immunoblot analysis of SDS cell extracts from wild-type (WT) and XRCC1KO hTERT RPE1 cells with late PARG inhibition (10 μM) for the indicated time points before harvesting at 45 min after 150 μM H2O2-treatment with the indicated antibodies (right). $N = 3$. Source data are provided as a Source Data file.

ADPrUb substrates, including their degradation, stabilisation or relocalization[35,36,53].

## Discussion

Although recent methodological advances are addressing long-standing challenges in ADPr research[16], specific and sensitive detection of all ADPr forms remains limited. Here, we extend our serine ADPr technology[18,20,21] by introducing a mono-ADPr blocking strategy. As a technology, serine ADPr offers several advantages inherent to the nature of the physiological reaction. First, PARP1 is strongly activated by broken DNA, making the reaction scalable to quantities suitable for antibody development[21]. Second, HPF1 not only confers serine specificity on PARP1, but also enables ADPr of peptides[14,21]. Third, the interplay between serine ADPr and canonical histone marks[54] can be harnessed to render the PARP1/HPF1 writer complex programmable, directing it to a specific serine by using phosphorylation as an unconventional protecting group[21]. Fourth, serine-ADPr is chemically stable with its the O-glycosidic linkage intact across a wide pH range and heat exposure up to 95 °C[17], minimising sample loss, a common issue with other forms of ADPr, particularly aspartate/glutamate

ADPr[17,49,52]. Fifth, PARG cleaves mono-ADPr from several residues[17,44] but not serine[21], allowing generation of defined serine mono- and poly-ADPr species for antibody testing (Figs. 3A, B and 4F)[18].

The development of sensitive and specific antibodies in this work has enabled a multifaceted analysis of ADPr in the context of XRCC1 deficiency, including the detection of low poly-ADPr levels, such as those in untreated RPE1 cells, that are not detectable with other tools. As demonstrated by Demin et al.[10], XRCC1 prevents excessive and prolonged PARP1 engagement with repair intermediates during BER, thereby acting as a PARP1 anti-trapper that ensures efficient, unobstructed DNA repair. In the absence of XRCC1, PARP1 becomes persistently associated with BER intermediates, leading to an early burst of PARP1 catalytic activity. This initial hyperactivation depletes NAD+, thereby limiting PARP1-mediated poly-ADPr at later time points[10]. Importantly, the ADPr dynamics in XRCC1-deficient cells differ markedly between MMS and H2O2, reflecting how XRCC1 loss interacts with the distinct repair pathways engaged by these agents. MMS generates alkylated bases that produce a high burden of BER intermediates, which accumulate in the absence of XRCC1 and promote early PARP1 trapping and hyperactivation[10], whereas H2O2 induces strand breaks

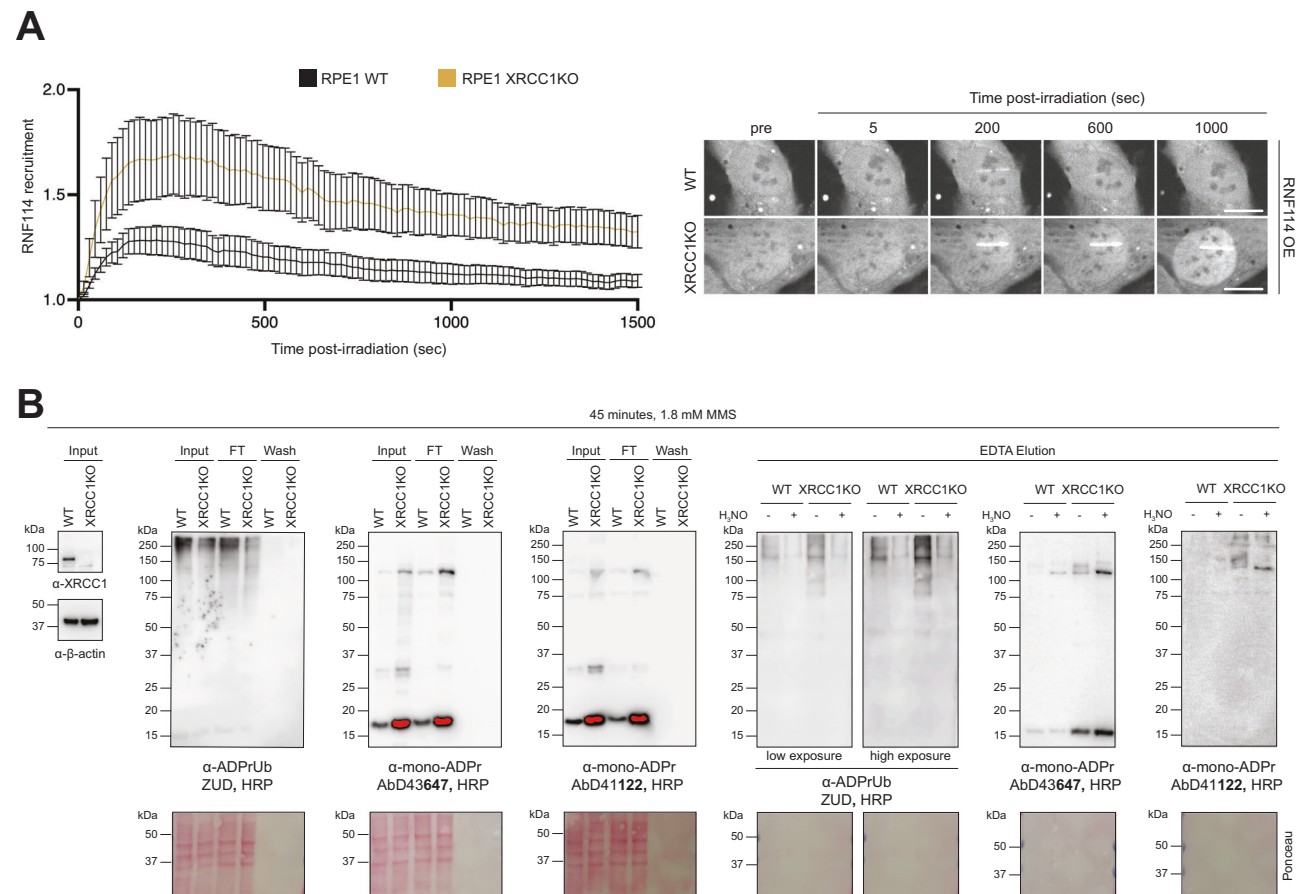

**Fig. 8 | XRCC1 deficiency increases cellular levels of DNA damage-induced ADP-ribosyl-ubiquitylation. A** Real-time live-cell detection of RNF114 in hTERT RPE1 WT or XRCC1KO cells. Representative recruitment kinetics (left) and confocal images (right). Error bars represent SEM. Scale bar, 10 µM. $N = 3$. **B** Immunoblot analysis of Streptavidin pulldown using biotin-coupled ZUD to enrich ADPrUb in 1.8 mM MMS treated hTERT RPE1 WT or XRCC1KO cells. Total cell lysates as Input, Elution performed with EDTA-containing non-denaturing buffer, to allow specific ADPrUb protein elution. Red colour represents a saturated signal. $N = 3$. Source data are provided as a Source Data file.

and oxidative lesions that enter BER through different entry points and with distinct processing kinetics. Although the precise mechanisms remain unresolved, these XRCC1-dependent differences in BER intermediate load and repair dynamics are consistent with the distinct ADPr responses we observe for MMS and $H_2O_2$. To our knowledge, no prior study has directly compared mono- and poly-ADPr dynamics across these two damage types in XRCC1-deficient cells, uncovering a stimulus-dependent divergence made apparent by the sensitivity of our antibody toolkit. Intriguingly, similar to our recent observations in ARH3 deficiency[33], the increase in mono-ADPr in XRCC1 knockout cells is accompanied by elevated ADP-ribosyl-ubiquitylation and recruitment of its reader, RNF114, to DNA lesions (Fig. 8). This raises the question of whether the accumulation of ADP-ribosyl-linked serine ubiquitylation[33] contributes to neurodegeneration in ARH3- and XRCC1-deficient patients.

While pan-ADPr antibodies developed by us and others[21,45] offer broad detection, they cannot distinguish between mono- and poly-ADPr, two distinct signalling events, even within pathways where they co-occur[17,18]. This lack of resolution can lead to misleading conclusions, particularly in the DNA damage response, where mono- and poly-ADPr represent the first and second wave of PARP1 signalling, respectively[18]. For example, the use of pan- and poly-ADPr sensors at laser-induced DNA lesions led to the conclusion that short ADPr oligomers constitute the sustained signal[55]. However, mono-ADPr specific sensors and antibodies revealed that the sustained phase of ADPr signalling can be more precisely attributed to

mono-ADPr[18]. In the context of XRCC1, our findings indicate that part of the pan-ADPr signal previously attributed solely to poly-ADPr[11] originates from mono-ADPr. In addition, continuous poly-ADPr formation even in the absence of exogenous DNA damage (Fig. 3D, E)[46] can mask mono-ADPr when using pan-ADPr antibodies, as shown for interferon signalling, a limitation overcome by mono-ADPr-specific antibodies[24]. Consequently, our toolbox allows for precise and sensitive detection of each modification, enabling a clearer understanding of their distinct contributions to signalling. In addition, our mass spectrometry method opens up the possibility of studying mono- and poly-ADPr at specific modification sites, potentially even in the context of ADPr-Ub, overcoming the obstacle of differentiating between mono- and poly-ADPr, which hindered studying PARP1-induced ADPr during replication[56].

Importantly, our mono-ADPr blocking strategy opens the door to guiding phage display selection toward other forms of ADPr that, within this framework, can be considered variations of mono-ADPr (Fig. 1A). This principle could enable the development of antibodies specific for ADP-ribosyl-ubiquitylation, greatly enhancing proteomic identification of ADPrUb sites by enabling peptide-level enrichment. While our mono-ADPr antibodies already play key roles in studying ADPrUb[31–33,36], the broader effort to develop specific tools for this composite modification has only just begun with the introduction of SpyTag-ZUD[33].

In conclusion, our mono-ADPr blocking strategy expands the toolkit for studying ADPr, particularly by generating poly-ADPr-

specific antibodies. We incorporated serine ADPr throughout the antibody generation process, from the use of precisely modified peptide as antigens to multiple validation steps and in-depth characterisation. The new antibodies revealed a mono-ADPr response to XRCC1 deficiency, along with its link to ADP-ribosyl-ubiquitylation. Our work underscores the importance of distinguishing between mono- and poly-ADPr and provides tools to explore them across diverse signalling pathways.

## Methods

### Cell Culture and drug treatments

U2OS cell lines were obtained, authenticated by STR profiling and confirmed mycoplasma-free by ATCC cell line authentication service. Cells were cultured in GlutaMAX-DMEM supplemented with 10 % bovine serum and 100 µ/ml penicillin/Streptomycin at 37 °C and 5 % $CO_2$. hTert RPE1 cell lines and primary fibroblasts with XRCC1 mutations[11] as well as control counterparts 1BR.3 (1BR) were generously provided by Dr. Keith Caldecott and confirmed mycoplasma free in in-house testing. Cells were cultured in GlutaMAX-DMEM supplemented with 10 % bovine serum and 100 U/ml penicillin/Streptomycin at 37 °C and 5 % CO2 with low-oxygen (3 %).

To induce DNA damage, the cell medium was aspirated and replaced with 37 °C complete DMEM containing 2 mM (U2OS) or 150 µM $H_2O_2$ (RPE1), 1.8 mM MMS or 15 µM CPT for indicated time points. For PARP inhibition (Olaparib, Cayman Chemical) cells were treated with 1 µM Olaparib in complete DMEM for 30 min prior to DNA damage treatment, followed by indicated $H_2O_2$ & 1 µM Olaparib treatment. For PARP1 specific inhibition, cells were treated with 10 nM AZD9574 (Palacaparib, MedChemExpress) 30 min prior to DNA damage treatment, followed by indicated $H_2O_2$ & 10 nM AZD9574 treatment. For PARG inhibition, cells were treated with 1 µM PARG inhibitor (PDD00017273, AdipoGen) 30 min prior to DNA damage treatment, followed by indicated H2O2 & 1 µM PARG inhibitor treatment.

For glucose starvation, cells were washed with phosphate-buffered saline (PBS) and incubated in glucose-free DMEM (L-glutamine) supplemented with 10% heat-inactivated fetal bovine serum (FBS) and 100 µ/mL penicillin/streptomycin for 3 h at 37 °C.

### Plasmids

Plasmid for mEGFP-WWE (encoding the WWE domain of RNF146) was kindly provided by Dr. Sebastien Huet (University of Rennes). pmCherry-macroD2 (encoding the macrodomain of macroD2) was kindly provided by Dr. Gyula Timinszky (Biological Research Centre, Szeged, Hungary). GFP-H1.4 Plasmid was kindly provided by Dr. Ivan Ahel (Sir William Dunn School of Pathology, Oxford, England).

### Antibody SpyCatcher coupling

All Bio-Rad antibodies have been obtained unconjugated in a monovalent format and were conjugated to BiSpyCatchers in-house leading to bivalent HRP conjugated or bivalent synthetic IgG or Biotin coupled antibody formats. In brief antibody stocks were diluted to 1 mg/ml in PBS and incubated with 1/10th the volume of unconjugated antibody with BiSpyCatcher for 1 h at RT, i.e., 10 µl of BiSpyCatcher was added to 100 µl of 1 mg/ml unconjugated antibody. Afterwards, they were aliquoted and stored at − 20 °C.

### Expression and purification of recombinant proteins

Proteins were expressed in *Escherichia coli* (*E. coli*) and purified essentially as previously described[57], but with the addition of a size-exclusion chromatography purification using a HiLoad 16/60 Superdex 75 column. PARP1 wild-type was expressed and purified as previously reported[57,58]. PARG was purified as described[59]. HPF1 was expressed and purified as reported[60].

### In vitro modification of proteins

Recombinant PARP1 protein (0.5 µM) was incubated with 2 mM NAD⁺ in the presence of HPF1 (5 µM). The reaction buffer contained 50 mM Tris-HCl, pH 8.0, 100 mM NaCl, 2 mM MgCl2 and 1 ng/µL sonicated DNA. Reactions were allowed to proceed for 30 min at RT, before being stopped by the addition of 1 µM Olaparib. To create mono- and poly-ADP-ribosylated protein samples were split, and one half was further incubated with 3 µM PARG for 1 h at RT and stopped by the addition of Laemmli buffer. Samples were boiled and run on an 8% Bis-Tris gel (Sigma-Aldrich).

For Dot-blotting the same reaction as above was performed without splitting the reaction. No loading buffer was added. A dilution series of the sample was performed using reaction buffer, and the same volumes (1 µl) of the probe was pipetted onto a dry nitrocellulose membrane. The membrane was left to dry and was proceeded with 5 % non-fat dried milk blocking and regular immunoblotting steps as described below.

### In vitro Modification to generate Poly-ADPr Peptides

PARP1 (PVEVVAPRGKSGAALSKKC), H3 (ARTKQTARKSTGGKAC) and H1.4 (KATGAATPKKSAKKTPKKGGK) peptide sequences were chosen based on Serine-ADPr abundances in cell-based studies. Peptide sequences were modified by adding a Cysteine for follow up Protein coupling. In separate in vitro reactions of 250 µl, peptides (1 µg/µl) were incubated in the presence of 0.1 µM PARP1, 5 µM HPF1 and 2 mM NAD⁺ for 6 h at RT, adding a final 2 mM NAD⁺ every 2 h to the reaction. Afterwards, the reaction was split in two and half it was incubated with 3 µM PARG for 1 h at RT. The reaction buffer contained 50 mM Tris-HCl, pH 8.0, 100 mM NaCl, 2 mM MgCl2 and 1 ng/µL sonicated DNA. Peptides were desalted and purified from recombinant proteins on in-house-manufactured SDB-XC StageTips. In brief. Stagetips were precleared using 100 % Acetone followed by isopropanol and spin until dry. Afterwards, Stagetips were activated using 100 % Methanol and spin until material still wet, followed by one wash in $H_2O$. Peptides were loaded in $H_2O$, followed by two washes in $H_2O$. Peptides were eluted using 40% ACN in 0.1% FA and afterwards freeze-dried. Reaction without PARG was resuspended and further ADP-modified in the presence of 1.2 µM PARP1 and 2 mM NAD⁺ for 6 h at RT, adding final 2 mM NAD⁺ every 2 h to the reaction. Again, the reaction buffer contained 50 mM Tris-HCl pH 8.0, 100 mM NaCl, 2 mM $MgCl_2$ and 1 ng/µL sonicated DNA. Afterwards, peptides were desalted und purified as before and freeze-dried.

### In vitro modification of mono-ADPr Peptides for Dot-Blot

PARP1 (APRGKSGAALSKKS(ph)KGQVGGK(Biotin)), H3 (1-21)(ARTKQTARKSTGGKAPRKQLAGGK(Biotin)), H3 (22-44) (ATKAARKSAPATGGVKKPHRYRPGGGK(Biotin)) and H1.4 (KATGAATPKKSAKKTPKKGKK(Biotin)) peptides, were modified for mono-ADPr as described above. In brief, peptides (1 µg/µl) were incubated in the presence of 0.1 µM PARP1, 5 µM HPF1, 3 µM PARG and 2 mM NAD⁺ for 6 h at RT, adding a final 2 mM NAD⁺ every 2 h to the reaction. The reaction buffer contained 50 mM Tris-HCl, pH 8.0, 100 mM NaCl, 2 mM $MgCl_2$ and 1 ng/µL sonicated DNA. Peptides were incubated with Avidin (Sigma-Aldrich, #A9275) 1:1 (w/w) for 5 min at RT. Coupled Peptides were directly pipette onto 0.2 µm Nitrocellulose membranes, left until dry and blocked with 5% non-fat milk and incubated with antibodies as described below.

### Dot-Blot with free-mono-ADPr and free-poly-ADPr

To analyze specificity of poly-ADPr specific antibodies, dot blotting was performed using Biotinylated ADPr (Jena Bioscience) and Biotin(Terminal)-Poly-ADPr (R&D Systems), which were prepared in four serial (1:2) dilutions in water. For each dot, 1 µL of prepared dilution was pipetted onto 0.2 µm Nitrocellulose membrane, left to dry around one hour at RT and blocked with 5% BSA in 1 x PBS. After blocking, HRP-

formats of our poly-ADPr antibodies (0.1 µg/mL) and pan-ADPr antibody (Cell Signalling Technology, 1:1000) were diluted in 1 % BSA in 1x PBS-T, and the membranes were incubated for 2 h at RT and mild shaking. Membranes were then washed three times for 5 min with PBS-T. Pan-antibody was additionally incubated with secondary rabbit-HRP antibody diluted in 1% BSA in 1x PBS-T (1:8000) for 1 h at RT and mild shaking with subsequent washing three times for 5 min with PBS-T. Membranes were developed at ChemiDoc using ECL Select Reagents.

### Inverted polarity visualisation technique for peptide-ADPr detection

Substrates and in vitro ADPr modified peptides were visualised as described before[54]. In brief, 3 µg of peptide were mixed with TBE sample buffer and loaded on 20 % TBE gel in TBE running buffer. Samples were run with inverted polarity for 90 min at 200 V. Gels were fixed with for 30 min in 10% Glutaraldehyde and stained with Imperial Protein Stain for 1 h after washing.

### Acid–Urea Gel Electrophoresis of ADP-ribosylated Peptides

Biotinylated poly-ADP-ribosylated histone H1 peptides (see section above) were analysed by acid–urea (AU) gel electrophoresis based on a published protocol[61], with modifications as detailed below. Where indicated, peptides were treated with recombinant PARG for 1 h at 37 °C to convert poly-ADP-ribosylation to mono-ADP-ribosylated forms. Samples were mixed with 2 × Gel Loading Buffer II (Invitrogen, # AM8546G), and up to 5 µg peptide per well was loaded.

AU gels (19:1 acrylamide:bisacrylamide) were cast at 1.5 mm thickness. Resolving gels contained 2.4 g urea, 4.5 ml 40 % acrylamide–bisacrylamide (19:1), 0.5 ml glacial acetic acid, and were filled up with water to 9.75 ml, supplemented with 200 µl 10% APS and 40 µl TEMED. Stacking gels contained 2.4 g urea, 1.25 ml 40% acrylamide–bisacrylamide (19:1), 0.25 ml glacial acetic acid, and were filled up with water to 4.75 ml, with 200 µl 10% APS and 40 µl TEMED. Before adding APS and TEMED, the resolving and stacking solution were pre warmed to 30 degrees for 30 min to ensure seamless polymerisation.

Gels were pre-run at 150 V with inverted polarity in 5% acetic acid, using equilibration loading buffer (8 M urea, 10 mM Tris pH 7.5, 10% PEG 6000) either overnight or until the current dropped below 30 mA (ca. 2 h). Samples were then run at 290 V with inverted polarity for up to 45 min in fresh 5% acetic acid. To visualise the running front, 1 µl methyl can be added to a free pocket.

For staining, gels were fixed in 10% glutaraldehyde for 30 min, washed 3 times with PBS, stained with Imperial Blue for 2 h, and destained overnight. For immunodetection, unfixed gels were transferred onto 0.2 µm PVDF-PSQ membranes in NuPAGE Transfer buffer with 15 % methanol at 40 V overnight at 4 °C. Membranes were cross-linked in 0.05 % glutaraldehyde for 20 min, washed in PBS three times, blocked in 5 % BSA for 1 h, and incubated with the indicated HRP-conjugated antibodies (0.0005 ug/ml) or Mono-/Poly-ADPr antibody (1 ug/ml; #89190S, Cell Signalling Technology) for 2 h RT followed by 1 h secondary antibody incubation for uncoupled Cell Signalling Antibody. Signals were detected by chemiluminescence using Cytiva's Amersham ECL Select Western Blotting Detection Reagent or Super-Signal West Atto Ultimate Sensitivity Substrate (Thermo Scientific).

### RNA isolation and enzymatic treatment

Total RNA was isolated using the Quick-RNA Miniprep Kit (Zymo Research) according to the manufacturer's instructions and quantified by NanoDrop spectrophotometry. For digestion controls, 5 µg of RNA was incubated in 50 µL of 1 × SSC buffer at 37 °C for 30 min with either 1 µL RNase A/T1 mix (2 mg/mL RNase A, 5000 µ/mL RNase T1; Thermo Fisher Scientific) + 1 µL RNase I (100 µ/µL; Thermo Fisher Scientific), or 1 µL Proteinase K (20 mg/mL). Digested samples and untreated controls were purified twice using the Monarch RNA Cleanup Kit (New England Biolabs) according to the manufacturer's instructions. Equal

amounts of RNA from all conditions (untreated and digestion controls) were used for dot blotting. For the detection of ADP-ribosylated RNA, RNA samples (100–2000 ng) were diluted in 200 µL of 20 × SSC buffer (3 M NaCl, 300 mM sodium citrate, pH 7.0) and spotted onto a nitrocellulose membrane using a Bio-Dot Microfiltration Apparatus (Bio-Rad) according to the manufacturer's instructions. Membranes were air-dried for 10 min and UV-crosslinked (254 nm, 125 mJ/cm²). Membranes were washed once with PBST (PBS + 0.1% (v/v) Tween-20) for 5 min at RT with gentle shaking for the removal of the unbound RNA, then blocked in 5% (w/v) non-fat milk in PBST for 1 h at RT. Membranes were incubated overnight at 4 °C with HRP-conjugated antibodies: AbD43647 or AbD41122 (both 0.1 µg/mL) in 5% milk–PBST. After three 10-min washes with PBST, membranes were developed using enhanced chemiluminescence (ECL) reagent. Signals were acquired using a ChemiDoc MP Imaging System (Bio-Rad).

### Antibody generation

Antibody generation was performed as described before[21]. In contrast to the previously described protocol, here mono-ADPr H3S10ADPr peptides with a C-terminal cysteine (modification protocol described above) were conjugated to the carrier proteins bovine serum albumin (BSA) and human transferrin (Trf) and used for blocking during the panning process. Mono-/Poly-H3S10ADPr peptide mix with a C-terminal cysteine (modification protocol described above) were conjugated to the carrier proteins bovine serum albumin (BSA) and human transferrin (Trf) and immobilised to Maxisorp plates. After selection rounds the antibody genes were subcloned as a pool into a vector for expression of a bivalent Fab format Fab-A-F. And after expression in *E. coli* cultures were lysed and crude extracts were tested in ELISA, using mono-ADPr H3S10ADPr peptide, mono-ADPr PARP1-S499ADPr peptide and mono-/poly-ADPr H3S10ADPr peptide mix for selection. Clones fulfilling criteria were sequenced, and the resulting unique antibodies were expressed and purified as previously described[62]. Purity and activity were subsequently tested by Coomassie-stained SDS-PAGE and indirect ELISA, respectively. All positive clones were validated using in vitro modified proteins and peptides, as well as with cell extracts in immunoblotting and fixed cells in immunofluorescence microscopy.

### Antibody affinity maturation

Antibody affinity maturation was performed similar as described before[18,63]. The antibodies AbD33205 and AbD33644 were derived from the Fab phage display library HuCAL PLATINUM[21,64].

For AbD33205 maturation leading to the successfully validated AbD41122 clone, we employed a selection strategy utilising a PARP1S499-mono-ADP-ribosylated peptides (APRGKS(ADPr) GAALSKKSKGQVGGK-biotin), to preserve - and potentially enhance - the broad mono-specificity of the parental antibody.

For AbD33644 maturation leading to the successfully validated AbD55568 and AbD55558 clones, we employed a selection strategy utilising the H3S10ADPr biotinylated peptide (ARTKQTARKS(ADPr) TGGKAPRKQLAGGK-biotin).

To identify successfully affinity-matured clones, a subset of candidates was evaluated by ELISA.

### Protein overexpression for histone H1.4 and live cell imaging

RPE1 cells were transfected with Plasmids described above, using X-fect (Takara) according to the manufacturer's protocol. In brief, 10:3 dilutions of Plasmid:Polymer were prepared in reaction buffer and incubated for 10 min. Protein expression was performed for 48 h before harvesting or imaging.

### Live-cell imaging

For live-cell imaging experiments, cells were seeded into m-Slide 8-well polymer-bottom chambered coverslip (Ibidi) or m-Dish 35 mm

polymer-bottom coverslip (Ibidi). For Hoechst presensitization, the culture medium was removed and replaced with fresh medium containing 0.3 mg/mL Hoechst 33342. Cells were incubated in this solution for 1 h at 37 °C. Immediately before imaging, the medium was exchanged for a $CO_2$-independent imaging medium (Molecular Probes Live Cell Imaging Solution (Invitrogen), supplemented with 20% fetal bovine serum (FBS), 2 mM glutamine, 100 μg/mL penicillin, and 100 μ/mL streptomycin).

Protein recruitment kinetics at laser-induced DNA damage sites were monitored as previously described[65]. Briefly, imaging was carried out using an Olympus SpinSR spinning disk confocal system equipped with a Yokogawa CSU-W1 spinning disk head (50 μm pinhole size), a UPLSAPO 100 ×/1.35 NA silicon-immersion objective, and a Hamamatsu ORCA-Flash4.0 sCMOS camera. Laser microirradiation was performed on Hoechst-presensitized cells along a 10 μm line across the nucleus using a continuous 405 nm laser at an intensity of 125–130 mW at the sample plane. Cells were maintained at 37 °C using a temperature-controlled chamber during imaging. Image sequences were analysed using ImageJ/FIJI or Olympus CellSense. The irradiated region and entire nucleus were manually segmented, and the mean fluorescence intensity within the irradiated area was background-subtracted. This value was then divided by the mean nuclear intensity to correct for imaging-induced photobleaching and subsequently normalised to the pre-damage signal.

### ADPrUb Immunoprecipitation
Cells were seeded in two big square dishes (500 cm² each)/condition and treated as described before. Cells were harvested in ice-cold PBS and centrifuged for 5 min, 500 x $g$ at 4 °C. Ice cold Lysis Buffer (20 mM HEPES, pH 7.0; 300 mM NaCl; 2/5 mM $MgCl_2$; 0.5 % NP40; 20 % Glycerol, 1:1000 benzonase (750 U/ml), 20 μm Olaparib, 2 μM ADP-HPD, EDTA-free Protease Inhibitor, PR619 1:1000) was added to the cell pellet and incubated for 1 h at 4 °C, end-to-end rotation. Lysates were cleared by centrifugation at 16,000 x $g$, 5 min at 4 °C. Protein concentration was measured using Pierce Dilution-Free BCA Protein Assay Kit (#A55864, ThermoScientific). Lysate volumes corresponding to 6 mg Protein were diluted at least twice using Dilution Buffer (20 mM HEPES pH 7.0; 0.5 % NP40; 20 μm Olaparib, 2 μM ADP-HPD, EDTA-free Protease Inhibitor, PR619 1:1000). 200 μg ZUD[33], was coupled to Biotin BiSpyCatcher2 (BioRad, #TZC002B) according to the manufacturer's protocol. Afterwards, ZUD-Biotin was incubated with prewashed Dynabeads MyOne C1 Streptavidin beads (ThermoFisher, #65001) and incubated for 1 h at 4 °C, end-to-end rotation. Coupled beads were washed twice using NP40 containing washing buffer (20 mM HEPES pH 7.0; 300 mM NaCl, 0.05% NP40). Half of the beads were used per condition and incubated with the diluted lysates for 1 h at 4 °C, end-to-end rotation. Beads were washed 3x using NP40 containing washing buffer, followed by 3x washed NP40 free washing Buffer (20 mM HEPES pH 7.0; 300 mM NaCl). ADPrUb modifed proteins were specifically eluted using 30 μl EDTA Elution Buffer (30 mM EDTA; 20 mM HEPES pH 7.0; 300 mM NaCl). Elutions were split in two, half of the elution was treated with $NH_2OH$ (1 M final concentration) for 2 h at RT. 1/10th of each sample was loaded for mono-ADPr antibodies, 9/10th of each sample was loaded for ZUD-HRP coupled detection. Samples were run on 4–12% Bis-Tris gels (mPage, Sigma-Aldrich) and run with precooled MES running buffer (Invitrogen). Gels were transferred on PVDF membranes (Amersham) for 110 min at 110 V on ice. Membranes were blocked in 5 % non-fat dried milk for 30 min at RT, followed by overnight incubation of Antibodies/Reagents at 4 °C. Membranes were washed with PBS-T 3 × 10 min before developing using ECL Select Western Blotting Detection Reagent (Amersham) or SuperSignal West Atto Ultimate Sensitivity Substrate (ThermoFisher).

### ADPr Immunoprecipitation
Cells were seeded in 10 cm cell culture dishes and treated with $H_2O_2$ as described above. Cells were harvested in ice-cold PBS, and centrifuged for 5 min, 500 xg at 4 °C. Afterwards, supernatant was aspirated and the cell pellet was resuspended in RIPA Buffer (50 mM HEPES, pH 7.0; 150 mM NaCl; 1 % NP40; 0.5 % Sodium deoxycholate; 0.1 % SDS). 1 μl benzonase (smDNase, 750U per sample was added and incubated for 10 min on ice. Samples were spin at max speed for 10 min and supernatant was transferred into new tube. The sample was diluted 1:3 with PBS containing ADP-HPD, Olaparib and Protease Inhibitors. 200 μg Lysate was incubated with 15 μg Antibody for 4 h at 4 °C end-to-end rotation. 15 μl prewashed Protein A (ThermoFisher) or high-capacity Streptavidin beads (ThermoFisher) were added to Lysate/Antibody mix and incubated for 1 h at RT. Beads were washed 3 x with PBS. Proteins were eluted using 4x LDS Buffer supplemented with DTT at RT second elution was performed with 4x LDS/DTT Buffer at 95 °C.

### Histone H1 isolation
Cells were seeded on big square dishes (500 cm²), treated as described above and harvested in ice-cold PBS and spun for 5 min, 500 x $g$ at 4 °C. Cell pellet was resuspended in 500 μl 0.1 M ice cold $H_2SO_4$ and incubated for 2 h on ice. Samples were cleared by centrifugation 2200 x $g$ for 20 min at 4 °C, resulting Pellet referred to as the $H_2SO_4$ pellet. 100 μl 70 % perchloric acid was added and incubated for 1 h on ice. Lysates were cleared by centrifugation 10,000 x $g$ for 30 min at 4 °C, resulting Pellet referred to as PCA pellet. 300 μl 100 % trichloroacetic acid (TCA) was added and incubated overnight at 4 °C, samples were cleared by centrifugation at max speed for 30 min, resulting Pellet referred to as TCA pellet, this pellet contains Histone H1. All pellets were washed with 2 washes in ice-cold acetone + 0.2% HCl followed by one wash in ice-cold acetone. Pellets were air dried and resuspended in $H_2O$, Protein concentration was measured using Pierce Dilution-Free BCA Protein Assay Kit (#A55864, ThermoScientific), and samples were loaded on SDS Page gels and continued as described below.

### $H_2SO_4$ Sample preparation
RPE1 cells were treated as indicated and lysed in $H_2SO_4$, by first harvesting cells in PBS and centrifuging at 500 x $g$, 5 min at 4 °C. Cell pellets were resuspended in ice-cold 0.1 M $H_2SO_4$ and incubated for 2 h at 4 °C. Lysates were centrifuged for 20 min at 2200 x $g$ at 4 °C, supernatant was transferred into a fresh tube and neutralised with Tris/HCl pH 7.5. Concentrations were measured using a Pierce Dilution-Free BCA Protein Assay Kit (#A55864, ThermoScientific). Immunoblotting was performed as described below.

### SDS Sample preparation and immunoblotting
U2OS and RPE1 cells were treated as indicated, lysed in SDS buffer (4 % SDS; 50 mM HEPES, pH 7.2, 150 mM NaCl, 5 mM MgCl2) and incubated for 5 min at RT with recombinant benzonase (smDNase, 750U per sample). Samples were centrifuged at max. speed for 10 min and the supernatant was transferred into a fresh tube. Sample concentration was measured using a Pierce Dilution-Free BCA Protein Assay Kit (#A55864, ThermoScientific). Samples were prepared with NuPAGE LDS sample buffer with a final concentration of 5 mM DTT (Sigma) and loaded on 4–12 % Bis-Tris gel (mPage, Sigma-Aldrich). After SDS-Page, the gels are transferred onto PVDF membranes (Amersham) using wet transfers at 90 mA overnight on ice. The precooled transfer buffer is 1x NuPAGE transfer buffer (Invitrogen), 20 % ethanol in water.

For immunoblotting, membranes were blocked in 5% non-fat dried milk before primary antibodies were added. Primary Antibodies (commercial antibodies and HRP-formats of Bio-Rad conjugated antibodies) were prepared in milk using concentrations found in figures. The membranes were incubated overnight at 4 °C before being washed in TBS-T (25 mM Tris-HCl pH 7.5, 150 mM NaCl, 0.05 % Tween-20) or PBS-T 3 × 10 min. Commercial antibodies were incubated with a

secondary antibody (Anti-mouse IgG HRP-conjugated secondary Amersham Cat# NA931V, Anti-rabbit IgG HRP-conjugated secondary Merck Cat# GENA934-1ML) for 1 h at RT, followed by 3 × 10 min TBS-T or PBS-T wash. All membranes were developed using ECL Select Western Blotting Detection Reagent (Amersham). Quantification was performed in Fiji/ImageJ using rolling-ball background subtraction (radius 100 px) and fixed-size ROIs. Mean grey values were normalised as indicated, and data from three independent experiments are presented as mean ± SEM.

## Immunofluorescence

U2OS and RPE1 cells were grown on sterile glass coverslips in 24-well plates. treated as indicated and fixed for 20 min with Methanol at −20 °C. The fixed cells were washed twice with PBS, and blocked for 1 h, RT with 5% Normal-Goat-Serum in PBS supplemented with 0.3% Triton. Primary antibodies (commercial antibodies and synthetic IgG formats of Bio-Rad conjugated antibodies) were diluted as indicated and incubated overnight at 4 °C in 1x PBS, 1% BSA and 0.3% Triton. Coverslips were washed in PBS 3×5 min before adding the secondary antibodies and DAPI, again in 1x PBS, 1% BSA and 0.3 % Triton for 1 h, RT protected from light. Coverslides were washed again in PBS 3 × 5 min before mounted on microscopy slides using Prolong Diamond Anti-fade Mountant (ThermoFisher). Cells were scanned for immunofluorescence using a Leica SP8-X inverted laser-scanning confocal microscope.

For high-throughput imaging, cells were grown in Ibidi 96-well glass-bottom plates and stained as described above. After the final wash, the cells were stored in PBS and protected from light. Cells were then imaged using a Leica Mica Microhub, confocal setting 40x objective. Data analysis was performed as described before[18]. In brief, raw files from microscopy were imported to Fiji. After splitting channels, the DAPI-based channel was processed to generate a mask using the following filters. The mask was used to define regions of interest (ROIs), which were transferred to the ADPr fluorescence channel and mean grey values were measured. For data representation, all experiments were normalised as indicated in the figure legends. Means +-SEM of three biological replicates were depicted in graph. All values were imported to GraphPad Prism and plotted in a grouped data table.

## ELISA

Following protocol was used to check specificity of antibody binding by ELISA: wells of the Pierce NeutrAvidin Coated Black 96-Well Plates (ThermoScientific, #15117) were first washed three times with 200 μL PBS-T (0.05 % Tween-20 in 1 x PBS), then 100 μL of 61 nM peptide antigen in 1xPBS was added to each well (each antigen in duplicates per antibody tested) and incubated 1 h at RT, with mild horizontal agitation. After 1 h, the peptide mix was removed and wells were blocked with 5% BSA in PBS-T for 1 h at RT, with mild horizontal agitation. Wells were washed three times with 200 μL PBS-T before primary antibodies in the monovalent format (2 μg/mL in 2.5 % BSA) were added to the wells, incubated for 2 h at RT, with mild horizontal agitation. Wells were then washed five times with 200 μL PBS-T before secondary antibody (HRP-conjugated goat anti-human IgG, diluted 1:5000 in 5% BSA in PBS-T) was added to the wells and incubated 1 h at RT, with mild horizontal agitation. Wells were again washed five times with 200 μL PBS-T. To develop the signal, 100 μL of QuantaBlu mix (QuantaBlu Fluorogenic Peroxidase Substrate, # 15169, mixed according to the manufacturer instructions) was added to each well, and the signal was immediately developed in a microplate reader Tecan at 37 °C (excitation 325 nm, emission 420 nm).

## MS/MS analysis of in vitro modified peptides of H3 and H1.4

For MS/MS measurements about 20 ng of in vitro modified peptides were injected into an Oribtrap Q-Exactive HF MS (Thermo Scientific) or Orbitrap Lumos MS (Thermo Scientific) instrument connected to an EASY-nLC 1200 (Thermo Scientific) chromatographic system. 20 cm (Orbitrap Q-Exactive HF MS) or 35 cm (Orbitrap Lumos) columns were packed in CoANN 75-μm fritless emitters (MSWil, TIP36007515-20-5) with Poroshell 120-EC C18 medium (2.7 μm particle diameter). 0.1 % FA in water (Buffer A) and 0.1% FA / 80% ACN in water (Buffer B) were used as running buffers for chromatography. The data was collected in positive mode. The Orbitrap Lumos was equipped with a FAIMS Pro Duo interface (Thermo Scientific). The compensation voltages of the FAIMS were set to −40, −50 and −60 V.

On the Orbitrap Lumos, the following 60 min gradient and method was applied: 0 % to 4 % Buffer B increase in 1 min and 4% - 31% Buffer B in 39 min. To wash the column, Buffer B was increased to 90% over 10 min, followed by an additional 10 min wash with 90% Buffer B. A Top20 method was used, collecting MS1 spectra from 400–1600 m/z. An AGC target of 1200000 and a maximum injection time of 50 ms was set. For MS2 scans, the maximum injection time was set to 200 ms and the AGC target to 50000. The MS2 scan started at 120 m/z to include adenine diagnostic ions.

On the Q-Exactive HF MS, the following 52 min gradient and method was applied for H3 peptides: 0% to 4% Buffer B increase in 1 min and 4–31% Buffer B in 39 min, followed by an increase of Buffer B to 50 % in 5 min. To wash the column, Buffer B was increased to 90 % over 5 min, followed by an additional 2 min wash with 90 % Buffer B. A Top5 method was used, collecting MS1 spectra from 400–1600 m/z. An AGC target of 3000000 and a maximum injection time of 100 ms was set. For MS2 scans the maximum injection time was set to 200 ms and the AGC target to 1000000. The MS2 scan started at 120 m/z to include adenine diagnostic ions.

On the Q-Exactive HF MS the following 52 min gradient and method was applied for H1.4 peptides: 0 % to 20 % Buffer B increase in 45 min, followed by an increase of Buffer B to 50 % in 10 min. To wash the column Buffer B was increased to 90 % over 2 min, followed by an additional 3 min wash with 90 % Buffer B. A Top5 method was used, collecting MS1 spectra from 400–1600 m/z. An AGC target of 3000000 and a maximum injection time of 100 ms was set. For MS2 scans, the maximum injection time was set to 300 ms and the AGC target to 1000000. The MS2 scan started at 120 m/z to include adenine diagnostic ions.

More details of the used methods can be accessed via the provided raw files.

## MaxQuant analysis of in vitro modified peptides

Raw files of MS/MS analysis of in vitro modified H3 and H1.4 peptides were analysed using MaxQuant[66] Version 2.4.12.0.

The following settings were used for H3 peptide raw files: The maximum number of variable modifications was set to 4, with methionine oxidation, mono-ADPr on serine, di-ADPr on serine, tri-ADPr on serine and N-terminal acetylation allowed as variable modifications. Carbamidomethylation of cysteines was set as a fixed modification. No digest was performed, but to ensure detection of the peptide, trypsin was set as a digestion enzyme, and the number of missed cleavages was set to 10. A fasta containing only the H3 peptide sequence was used for the search. The minimum peptide length was set to 5, and a maximum peptide mass of 7000 was set. To also collect low-quality annotated spectra the minimum score for modified peptides and the minimum delta score for modified peptides were set to 0. The second peptide search was enabled. Match between runs was also enabled with default settings. The di-ADPr (H42 O26 P4 C30 N10) and tri-ADPr (H63 O39 P6 C45 N15) modification was set up in MaxQuant with neutral losses corresponding to AMP, Adenosine, Adenine, ADP, ADPr, DiADPr-AMP (H28 O19 P3 C20 N5) and DiADPr-ADP (H27 O16 C20 N5 P2). The diagnostic ions described in Supplementary Fig. 2 were included in the modifications. The resulting spectra were manually inspected for the presence of mono-ADPr diagnostic ions.

Three different searches with the following settings were used for H1.4 peptide raw files:

The maximum number of variable modifications was set to 4, with methionine oxidation and N-terminal acetylation always included as variable modifications. As a third modification either mono-ADPr on serine or di-ADPr on serine or tri-ADPr on serine were allowed as variable modifications. Carbamidomethylation of cysteines was set as a fixed modification. No digest was performed, but to ensure detection of the peptide, trypsin was set as a digestion enzyme, and the number of missed cleavages was set to 10. A fasta containing only the H1.4 peptide sequence was used for the search. The minimum peptide length was set to 7, and a maximum peptide mass of 7000 was set. Identification settings were left unchanged. The second peptide search was enabled.

More details on the MaxQuant searches can be accessed via the mqpar file provided in the PXD repository.

### Inspection of raw files for the presence of poly-ADPr diagnostic ions

Raw files were inspected for the presence of diagnostic ions using Freestyle. A layout in Freestyle was created using the diagnostic ions reported in Supplementary Fig. 2B. 5 ppm was set as a mass tolerance for each diagnostic ion.

### MaxQuant analysis of a public data set to identify di-ADPr sites

Raw files of the proteomic data set PXD023835[42] were searched for the presence of di-ADPr peptides. The raw files used for the search are labelled "20190116_LUMOS_LC4_SCL-IAH_AR_C_AF/AB1/AB2_RX_FX" in the original repository. Note that the file name either contains AF, AB1 or AB2, R values range from 1 to 4 and F values from 0 to 3, resulting in e.g., 20190116_LUMOS_LC4_SCL-IAH_AR_C_AF _R1_F0. The raw files correspond to the control group of an experiment in which ADP-ribosylated peptides were enriched using different reagents (AB1, AB2, AF) described in Fig. 2A of the original publication[42]. MaxQuant version 2.4.12.0 was used to analyze the data. The raw files were grouped in experiments and replicates, e.g., AB11 for AB1 Replicate 1, and fractions were assigned according to the F value in the file name. 3 variable modifications were allowed per peptide. Two searches were performed, one allowed Methionine oxidation, N-terminal acetylation and mono-ADPr on S,D,E,K,C,Y,T,H,R residues, and the second one allowed additional di-ADPr on the same amino acids. Carbamidomethylation was set as a fixed modification. Trypsin was set as the digestion enzyme with 3 maximum missed cleavages. For the first search, only methionine and N-terminal acetylation were allowed as separate variable modifications. As a fasta, the human proteome (UP000005640_9606) was set. Minimum peptide length was set to 7 and maximum peptide mass to 4600. To also collect low- quality annotated spectra, the minimum score for modified peptides and the minimum delta score for modified peptides were set to 0. The second peptide search was enabled. Match between runs was also enabled with default settings.

Di-ADPr sites reported in Fig. 2B–D as well as in Supplementary Fig. 2C were filtered to have a score > 40, delta score > 20 and a localisation probability > 0.8. The resulting spectra were visually inspected to check if the modification sites were covered by the sequence ions and if diagnostic ions were present. Note that MaxQuant did not annotate any diagnostic ions in EThcD spectra. Annotated Di-ADPr spectra were checked in MaxQuant and Freestyle for the presence of diagnostic ions, which are indicated in the respective figures (Fig. 2 and Supplementary Fig. 2).

More details on the MaxQuant searches can be accessed via the mqpar file provided in the PXD repository.

### Reporting summary

Further information on research design is available in the Nature Portfolio Reporting Summary linked to this article.

## Data availability

The antibodies AbD64138 and AbD41122 generated in this study are available through Bio-Rad Laboratories. Uncoupled Fab-SpyTag (monovalent "purified" format): AbD64138ad (Bio-Rad Catalogue # TZA0117): AbD41122ad (Bio-Rad Catalogue # TZA0118). Fab-SpyTag coupled to HRP-conjugated BiCatcher2 (bivalent HRP format): AbD64138pap (Bio-Rad Catalogue # TZA0117P). The other antibodies can be obtained in this format via conjugation to BiCatcher2:HRP (Bio-Rad Catalogue # TZC002P).

Fab-SpyTag coupled to rabbit IgG FcCatchers (IgG-like format): the antibodies can be obtained in this format via conjugation to rbIgG-FcSpyCatcher3 (Bio-Rad Catalogue # TZC013). Material requests should be addressed to the corresponding author.

The mass spectrometry proteomics data have been deposited to the ProteomeXchange Consortium via the PRIDE[67] partner repository with the dataset identifier PXD066208. Source data are provided in this paper.

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

## Acknowledgements
We thank the Caldecott laboratory (University of Sussex) for XRCC1 KO cells, controls, and patient fibroblasts; the Huet laboratory (Université Rennes) for GFP-WWE; the Timinszky laboratory (Biological Research Centre) for mCherry-MacroD2; and the Ahel laboratory (University of Oxford) for GFP-H1.4. We thank the MPI-AGE Proteomics and Imaging Facilities for support and the members of the Matić laboratory for discussions. This work was funded by the Max Planck Society; the DFG (Excellence Strategy - EXC 2030 – 390661388); and the European Research Council (ERC-CoG-864117) to I.M. The Cologne Graduate School of Ageing Research supported A.K. and C.K.

## Author contributions
I.M. conceived and supervised the project. I.M. and H.D. wrote the manuscript with input from all authors. H.D. performed cell culture, live-cell imaging, antibody generation and immunofluorescence. H.D. and M.D.P. performed immunoblotting and immunoprecipitations. H.D. and M.M. validated antibodies. A.K. performed proteomics experiments. C.K. and I.H. performed RNA experiments.

## Funding

## Competing interests
I.M., H.D. and M.M. declare competing financial interests related to licensed antibodies (AbD33205, AbD43647, AbD41122, AbD55568 and AbD64138) marketed by Bio-Rad Laboratories. I.M., A.K. and M.D.P. are inventors on a pending patent related to the ZUD reagent. The remaining authors declare no competing interests.
