## [Transparent Peer Review file · Nature Communications]

Versatile and sensitive detection of mono- and poly(ADP-ribosyl)ation reveals XRCC1-dependent remodelling of PARP1 signalling

Corresponding Author: Dr Ivan Matic

Version 0:

Reviewer comments:

Reviewer #1

(Remarks to the Author)

In this study, Dauben et al present a large body of data that addresses a number of loosely related topics that are of current interest in the ADP-ribosylation field. These include a) the development of new and improved antibody-like reagents for the specific detection of both mono-ADPr and poly-ADPr, b) the identification of diagnostic ions that allow the discrimination between mono-ADPr and poly-ADPr modifications by mass spectrometry, c) evidence that loss of XRCC1 impacts DNA damage-induced dynamics of PARP1-dependent mono-ADP-ribosylation, poly-ADP-ribosylation and ADP-ribosyl ubiquitylation. While the overall quality of the data is excellent, and the new reagents and methodologies described here will certainly influence further studies in the field, the study is somewhat narrowly focused on a mixture of methodological advancements, so my main two main suggestions focus on expanding the biological novelty of the manuscript.

1. Although the data in Figs 5-7 highlight the usefulness of using improved ADPr detection tools, the data on effects of XRCC1 loss on mono-ADPr, poly-ADPr and ADPr-Ub is very descriptive, without much new biological insight. While the experiments are carefully performed and demonstrate the importance of time course analyses for the study of DNA damage-induced ADPr dynamics, the findings that PARP1 activity is higher in XRCC1 KO cells and that the timing of mono vs poly-ADPr differs between different DNA damaging agents (both in WT and XRCC1 KO cells) are not particularly surprising. I strongly suggest a thorough discussion of how XRCC1 KO could be causing these differences in light of current literature, particularly ref. 8 (Demin et al Mol Cell 2021), which suggested that in MMS-treated XRCC1 KO cells, the excess PARP1 activity at early time points leads to NAD⁺ depletion, restricting PARP1 activity at later time points.

2. Related to the above point, an important experiment would be to determine the impact of PARG inhibition on the formation of excess mono-ADPr in XRCC1 KO cells. This would allow a distinction between a possible role of XRCC1 in influencing direct PARP1-dependent mono-ADP-ribosylation vs XRCC1 affecting the formation of mono-ADPr as a result of a potential impact on PARG-dependent poly-ADPr hydrolysis.

Minor comments:

3. Sup. Fig 2 would be a little easier to interpret if the diagnostic ions in panels A and B were exactly the same and displayed in the same order. Also, it would be helpful to add an extra column in table Sup. Fig 2B indicating which ions are present for both modifications, and which are specific for poly-ADPr.
4. The fairly frequent formation of adenosine (both from mono and poly-ADPr) indicates that cleavage at the glycosidic bond is frequent during MS fragmentation. Have the authors considered looking for other ions in which the adenosine moiety is missing?
5. The resolution of some of the MS spectra in the files I received was insufficient to properly read the y- and b-series fragment numbering on the peptide backbones at the top of the spectra (particularly Fig. 2), or to clearly distinguish between (colored) fragment ion peak heights and the lines (are they dotted?) used to label them.
6. Many of the western blot panels are cropped to a limited number of samples and then displayed side-by-side for

comparison. This is of course necessary when different ADPr detection reagents are used (Fig. 3, 4 and Sup. Fig.4), but hampers adequate comparison between samples during a time course (Fig. 5B and Sup. Fig 5D, E).

7. It is unclear what the “nuclear intensity” values used to plot microscopy quantifications represent. How were pixel intensities normalized to the arbitrary units in the ca. 1-100 range? Was background subtraction performed? Also, it seems that mean \pm SEM were determined as if each nucleus was an independent replicate (n is as high as 80 in some panels). This is incorrect. Instead, the values for all quantified nuclei for each biological replicate should be averaged, so that a mean intensity value for each biological replicate is obtained. These mean values per replicate are then used to derive a mean \pm SEM for the experiment, with n=3.

Typos:

- Fig 1A, top panel – “Stragety” should be “strategy”
- Fig. 2A, zoomed-in area of tri-ADPr peptide - ion 889.1315 label should be “AMP+ADPr”
- Line 157 – “AMP+ADP” cannot be correct. Maybe “ADPr+ADP”?

Reviewer #2

(Remarks to the Author)

This work reports the development of synthetic antibodies specific for poly(ADP-ribose) using chemoenzymatically prepared peptides with varied numbers of ADP-ribose and affinity maturation of synthetic antibodies specific for mono-ADP-ribose. Then, authors used selected antibodies reagents for poly(ADP-ribose) and mono-ADP-ribose to study ADP-ribosylation derived from PARP1 in XRCC1-deficient cells. Due to technical difficulties in preparing antigenic peptides carrying poly(ADP-ribose), authors employed a different blocking strategy by utilizing peptides with mono-ADP-ribose. This way allowed identification of new antibody reagents with improved sensitivity. Through phage display-aided selection, an antibody recognizing mono-ADP-ribosylated peptides in high specificity was discovered, showing different recognition patterns from a previously published antibody for mono-ADP-ribosylation. Combining the developed antibodies for poly(ADP-ribose) and mono-ADP-ribose, including ones reported in this manuscript and one from a previous study (abd43647), authors revealed time-dependent mono- and poly-ADP-ribosylation in wild-type and XRCC1 KO cells, and an increase of ADP-ribosyl-ubiquitylation in XRCC1 KO cells upon DNA damage. The significant efforts by authors in developing and affinity maturing antibody reagents for different forms of ADP-ribosylation are commendable. But the overall technical advances in generating ADP-ribosylation-specific reagents seem incremental compared with previously published studies. While the antibody specific for poly(ADP-ribose) showed higher sensitivity, new knowledge or information revealed exclusively by this reagent is little. Likewise, a new antibody more specific for mono-ADP-ribosylation was discovered from affinity maturation, but a previously published antibody with comparable affinity and specificity could possibly address the same questions. In addition to the technical novelty and application of the reported antibodies, the findings from XRCC1 KO cells about the ADP-ribosylation dynamics and the ADP-ribosyl-ubiquitylation provide limited insights into the functions and molecular mechanisms of mono- and poly-ADP-ribosylation and composite modification related to XRCC1. Only ADP-ribosylation dynamics under different DNA damaging conditions and increased ADP-ribosyl-ubiquitylation were shown. Functional and mechanistic understandings from these observations would be critical to support the essential roles of the reported toolbox in studying XRCC1-deficiency driven ADP-ribosylation. Given the modest technical innovation, already published antibody reagents with comparable affinity and specificity, and lack of functional and mechanistic insights from the XRCC1 deficiency-related findings, this manuscript is not recommended for publication in Nature Communications.

Reviewer #3

(Remarks to the Author)

The new study by Dauben et al. makes an important contribution to the protein ADP-ribosylation field, particularly in terms of providing new tools for studying this modification.

Regarding the tools, the manuscript primarily describes a strategy for selecting sensitive poly(ADP-ribose) antibodies; the development and characterisation of these antibodies for both immunoblotting and cell immunostaining applications; and the affinity-maturation of some previously developed site-specific and general mono(ADP-ribose) antibodies. The new anti-poly antibodies do appear particularly sensitive and specific.

In terms of biological insight, the authors apply some of these tools to show that cellular XRCC1 deficiency results in an increase in different ADP-ribosylation types including poly(ADP-ribosyl)ation, mono(ADP-ribosyl)ation, and a composite ubiquitin-mono(ADP-ribosyl) modification, the last mentioned possibly accounting for the increased accumulation of RNF114 (the reader of the composite modification in question) at the sites of laser-induced damage.

The study also contains further interesting observations, such as the somewhat different spatial distribution of mono- vs poly(ADP-ribose) signals in cells (in Suppl. Fig. 3D - by the way, can the extent to which these signals are “punctate” or not be quantified?) or - this is particularly interesting - the identification of a signature di(ADP-ribose) MS fragment that is specific for poly(ADP-ribosyl)ation and seems to have confounded previous site identification efforts by mimicking a double mono(ADP-ribose).

It is a rigorous and interesting study and I don't have any major points of criticism. Here are some minor points that could be taken into account to further improve it:

1) Are the new poly(ADP-ribose) antibodies specific for Ser-attached poly(ADP-ribose), or are they attachment-independent? Would they recognise Glu/Asp-linked poly(ADP-ribose)? Would they recognise free poly(ADP-ribose)? Could the authors discuss insights they might already have at their disposal that suggest answers to these questions? Would it take a lot of effort to explore this question further experimentally? I think it would be useful to at least discuss it. Similarly, do the authors think these antibodies might unequally detect chains differing in length or branching status?

2) In terms of citations, the selection of references is generally relevant and complete, but the authors could cite the 1960s studies they mention in line 39. The statement "Dysregulation of ADPr is implicated in various diseases, including cancer and neurodegeneration, ..." in lines 40-41 could also do with a citation supporting these claims. For the composite Ub-ADP-ribose modification in line 94, the Zhu et al. Sci Adv paper could be referred to alongside the cited cellular studies.

3) The sentence in lines 55-58 ("A prime example...") is a bit convoluted; the authors could consider simplifying it and perhaps splitting it in two.

4) I appreciate the fact that immunoblots are generally accompanied by Ponceau S stain images serving as loading controls, but these are sometimes a bit dark and low in contrast, especially when printed out - perhaps worth checking and improving their clarity.

5) In lines 283-285, a background signal that can be eliminated with sulphuric acid is mentioned. Could the authors comment on what this signal might be and why it gets eliminated with sulphuric acid?

6) Could the authors explain in clearer words the term "antibody leakage" used in line 300?

7) In line 332, could the authors clarify that they used both Olaparib and the more PARP1-specific AZD9574 to establish this conclusion?

8) The procedure involving perchloric acid enrichment and mutation used to identify H1 as one of the bands could be briefly explained in lines 337-338. This fragment contrasts with otherwise more thorough explanations of the approaches applied.

9) Lines 357-359: Is the more spatially heterogeneous distribution of poly(ADP-ribose) signal across cells due to the signal having already disappeared in some cells, or was it not there in some of them in the first place? Is it possible to distinguish between the two? What do the authors consider more likely?

10) The sentence "This raises the question whether the toxic accumulation of ADP-ribosyl-linked serine ubiquitylation contributes..." sounds as if it was known that ADP-ribosyl-linked Ub was toxic, and the remaining question was whether it contributed or not to these particular symptoms. However, the accumulation of this composite modification should only be called toxic if it does cause these symptoms and this is an open question as the authors admit. In short, perhaps it's better not to call it "toxic" until proven.

11) Line 552: space in E. coli missing. Also, some spaces between numbers and units are missing in Methods.

13) In Fig. 1A, it looks a bit as if the circle representing ADP-ribosyl was attached next to a serine ("S") rather than to a serine. Maybe it is just me.

14) Overall, the figures are high quality and clear and the legends are exhaustive, providing information on replicability, number of cells etc. The Methods description seems exhaustive.

15) The authors could consider changing poly-ADP-ribosylation etc. to poly(ADP-ribosyl)ation - but I guess it is their conscious choice to use the hyphenated form, which I respect.

Version 1:

Reviewer comments:

Reviewer #1

(Remarks to the Author)

In this revised version, Dauben et al addressed my concerns, including the addition of a new main figure (Fig. 7) exploring the effects of PARG inhibition on ADPr dynamics in XRCC1 KO cells, as suggested. As a result, the revised manuscript is much improved and I recommend publication. However, some minor issues still require some attention:

Sup Fig. 2A: The chemical structure for adenosine is missing on the left side of the figure, causing the chemical structures for most molecules to be misaligned with their respective profiles/labelling.

Fig. 4C: the minus (-) and plus (+) signs are presumably swapped, as glucose starvation implies lack of glucose.

Reviewer #2

(Remarks to the Author)

The new experiments included in the revised manuscript showed some new biological insights using the developed antibody reagents. However, a few concerns still need to be addressed to demonstrate the superiority of these antibodies over other established ones.

1. Fig. 3B tried to evaluate affinity among various anti-poly-ADPr reagents for poly-ADPr-PARP1 at varied concentrations. The reported two synthetic antibodies were labeled with HRP molecules, but other commercial ones carrying Fc from different species would need the use of distinct secondary antibodies for detection. This experimental design cannot ensure fair comparison of their affinity and justify correlations of high signal intensities with sensitivities of primary antibody reagents.
2. In Fig. 3A-E, 4F, and 7A, authors claimed significantly higher signals from the reported antibodies. To support these claims, quantitative analysis of the immunoblots with at least three independent replicates would be required.
3. In Fig. 4C, the authors claimed that AbD41122 allows to detect mono-ADPr on RNA molecules. How does this affinity-matured antibody compare with other commercial ones like CST D9P7Z, MABE1016, and MABE1076 for recognition of mono-ADP ribosylated RNA?
4. Lines 238 and 307, olaparib can inhibit PARP1 and other PARPs.
5. The glucose label for Fig. 4C is confusing. Should be glucose starvation.

Reviewer #3

(Remarks to the Author)

I thank the Authors for addressing all my comments, as well as many of those raised by other reviewers.

I particularly appreciate the new experiments exploring the specificity and sensitivity of the anti-poly antibodies, which I do think will make very useful tools. I now understand the challenges with quantifying the 'punctate' nature of the observed ADPr signals in cells, and with testing branchpoints, and I thank the Authors for explaining those. I also understand the challenge with Ponceau stain brightness, when little total protein is loaded; this is a common technical limitation.

Given the fact that, to me, the main value of the article lies in its methodological aspects (both in terms of antibodies and MS insights), I wonder whether the authors could still consider a different article title - one that does not highlight primarily the, somewhat secondary, biological insights, but rather both the antibodies and MS insights, presenting these as tools for monitoring PAR and MAR in different contexts. But this is just a gentle suggestion.

Overall, I think this manuscript is ready for publication and will be useful to the community. Congratulations.

Open Access This Peer Review File is licensed under a Creative Commons Attribution 4.0 International License, which permits use, sharing, adaptation, distribution and reproduction in any medium or format, as long as you give appropriate credit to the original author(s) and the source, provide a link to the Creative Commons license, and indicate if changes were

made.

Thursday, December 18, 2025

Point-by-point answer to Reviewers:

NCOMMS-25-57878 (A Versatile Toolbox Reveals XRCC1 Deficiency–Driven Reshaping of the PARP1 Mono-ADP-Ribosylation Wave)

Summary:

We would like to thank the editor and the reviewers for their time in evaluating our work and for their helpful suggestions. We are encouraged by the many positive comments, and we also appreciate the specific and broader concerns that were raised. In response to these comments, we have performed substantial additional experiments and made significant revisions to the manuscript.

Briefly, we have obtained four main sets of additional data:

- A new main figure analyzing the impact of PARG inhibition on the formation of excess mono-ADPr in XRCC1 KO cells
- Additional experiments demonstrating key advantages of the new tools over existing reagents: (a) Dramatic improvement in RNA ADPr detection by the new mono-ADPr antibody (data from a separate collaborative project and incorporated here); (b) detection of extremely low levels of PARP1-dependent poly-ADPr in unperturbed RPE1 cells, a signal not detectable with existing tools
- Extensive additional characterization of the poly-ADPr antibodies using a range of poly-ADPr forms (free chains, short oligomers and Asp-/Glu-linked) to demonstrate broad substrate recognition of the new tools.
- Strengthening of specific experiments in response to the reviewers' concerns

Further, we have carefully revised the text throughout in response to the reviewers' critique. We believe that by following reviewers' suggestions and incorporating considerable new results and clarifications, we have significantly improved the manuscript and addressed the concerns raised. We hope that the revised version will now be suitable for publication in *Nature Communications*. Please find below a detailed point-by-point answer to the reviewers' comments.

Table: Summary of Changes To figures

Previous version	Revised Version	
Fig. 1	Fig. 1	
Fig. 2	Fig. 2	Revised
Fig. 3	Fig. 3	Revised
Fig. 4	Fig. 4	Revised
Fig.5	Fig.5	Revised
Fig. 6	Fig. 6	Revised
	Fig. 7	New
Fig. 7	Fig 8.	
-	Fig. 2	
	Fig. 2a	Revised
	Fig. 2b	
	Fig. 2c	Revised
	Fig. 2d	Revised
-	Fig. 3	
	Fig. 3a	
	Fig. 3b	
	Fig. 3c	New
	Fig. 3d	Revised
	Fig. 3e	New
	Fig. 3f	Revised
-	Fig. 4	
	Fig. 4a	
	Fig. 4b	
	Fig. 4c	New
	Fig. 4d	Revised
	Fig. 4e	Revised
	Fig. 4f	
	Fig. 4g	
-	Fig. 5	
	Fig. 5a	
	Fig. 5b	Revised
	Fig. 5c	
	Fig. 5d	
-	Fig. 6	
	Fig. 6a	
	Fig. 6b	Revised
	Fig. 6c	
	Fig. 6d	
-	Fig. 7	New
	Fig. 7a	New
	Fig. 7b	New
-	Fig. 8	
	Fig. 8a	
	Fig. 8b	

Previous version	Revised Version	
Fig. S1	SupplementaryFig.1	Revised
Fig. S2	SupplementaryFig.2	Revised
Fig. S3	SupplementaryFig.3	Revised
Fig. S4	SupplementaryFig.4	Revised
Fig. S5	SupplementaryFig.5	Revised
Fig. S6	SupplementaryFig.6	
-	SupplementaryFig.1	
	SupplementaryFig.1a	
	SupplementaryFig.1b	
	SupplementaryFig.1c	Revised
	SupplementaryFig.1d	Revised
-	SupplementaryFig.2	
	SupplementaryFig.2a	Revised
	SupplementaryFig.2b	Revised
	SupplementaryFig.2c	Revised
-	SupplementaryFig.3	
	SupplementaryFig.3a	
	SupplementaryFig.3b	New
	SupplementaryFig.3c	New
	SupplementaryFig.3d	New
	SupplementaryFig.3e	Revised
	SupplementaryFig.3f	Revised
	SupplementaryFig.3g	Revised
-	SupplementaryFig.4	
	SupplementaryFig.4a	New
	SupplementaryFig.4b	Revised
	SupplementaryFig.4c	
	SupplementaryFig.4d	Revised
	SupplementaryFig.4e	
	SupplementaryFig.4f	
	SupplementaryFig.4g	
-	SupplementaryFig.5	
	SupplementaryFig.5a	
	SupplementaryFig.5b	
	SupplementaryFig.5c	
	SupplementaryFig.5d	Revised
	SupplementaryFig.5e	Revised
	SupplementaryFig.5f	Revised
	SupplementaryFig.5g	
-	SupplementaryFig.6	
	SupplementaryFig.6a	
	SupplementaryFig.6b	
	SupplementaryFig.6c	
	SupplementaryFig.6d	

Reviewer #1 (Remarks to the Author):

In this study, Dauben et al present a large body of data that addresses a number of loosely related topics that are of current interest in the ADP-ribosylation field. These include a) the development of new and improved antibody-like reagents for the specific detection of both mono-ADPr and poly-ADPr, b) the identification of diagnostic ions that allow the discrimination between mono-ADPr and poly-ADPr modifications by mass spectrometry, c) evidence that loss of XRCC1 impacts DNA damage-induced dynamics of PARP1-dependent mono-ADP-ribosylation, poly-ADP-ribosylation and ADP-ribosyl ubiquitylation. While the overall quality of the data is excellent, and the new reagents and methodologies described here will certainly influence further studies in the field, the study is somewhat narrowly focused on a mixture of methodological advancements, so my main two main suggestions focus on expanding the biological novelty of the manuscript.

We thank the reviewer for their thoughtful and constructive evaluation of our study. We greatly appreciate their acknowledgment that our work provides extensive experimental evidence covering multiple aspects of ADPr research that are timely and relevant to the field. We are also grateful for their recognition of the excellent quality of our data and for highlighting that our reagents and methodological advances are likely to stimulate and support future studies in the field.

We have carefully followed the reviewer's suggestions to expand the biological novelty of the manuscript. While the central aim of our study is the development of tools and methodologies, we have revised the manuscript to more fully integrate our findings on XRCC1 loss and ADPr dynamics with the current literature, and performed a new set of experiments involving inhibition of PARG and PARP1, as detailed below.

While we have thoroughly addressed the reviewer's two main points, as detailed below, we would like to clarify that the manuscript is best viewed in the *Resource/Technology* format. This could not be explicitly indicated at submission, as *Nature Communications* publishes all papers under a single "Article" format. From this perspective, and similarly to our *Bonfiglio et al. Cell 2020* study published in the *Resource* format, Figures 5–8 should primarily be viewed as a proof-of-principle demonstration that our new tools and methodologies will enable future investigation of key questions across different biological processes - a defining feature of the *Resource/Technology* format.

Validation of the newly developed tools and methodologies is a central aspect of this format. In this regard, we had already included extensive validation experiments in the original submission, as also acknowledged by the reviewer, and have now substantially strengthened this aspect with additional sets of experiments, as detailed in our responses to the specific points below.

1. Although the data in Figs 5-7 highlight the usefulness of using improved ADPr detection tools, the data on effects of XRCC1 loss on mono-ADPr, poly-ADPr and ADPr-Ub is very descriptive, without much new biological insight. While the experiments are carefully performed and demonstrate the importance of time course analyses for the study of DNA damage-induced ADPr dynamics, the findings that PARP1 activity is higher in XRCC1 KO cells and that the timing of mono vs poly-ADPr differs between different DNA damaging agents (both in WT and XRCC1 KO cells) are not particularly surprising. I strongly suggest a thorough discussion of how XRCC1 KO could be causing these differences in light of current literature, particularly ref. 8 (Demin et al Mol Cell 2021), which suggested that in MMS-treated XRCC1 KO cells, the excess PARP1 activity at early time points leads to NAD⁺ depletion, restricting PARP1 activity at later time points.

We thank the reviewer for recognizing that our experiments were carefully performed and illustrate the importance of time-resolved analyses for understanding DNA damage-induced ADPr dynamics. While we fully agree that elevated PARP1 activity in XRCC1 KO cells is not surprising, we would like to highlight that XRCC1 deficiency has never been examined in the context of both PARP1 signaling waves. This is particularly important given the growing recognition of the mono-ADPr wave, which has not previously been considered in relation to XRCC1 deficiency, and its role in generating ADPr-linked ubiquitylation, a major emerging focus in the field.

We appreciate this insightful comment, which prompted us to revise several sections of the manuscript to better emphasize the unexpected aspects of our findings and to more clearly discuss them in the context of the specific influence of XRCC1 on PARP1 signaling. In particular, we have followed the reviewer's suggestion to more thoroughly interpret our results in the context of recent evidence that XRCC1 deficiency limits PARP1 activity at later time points, as reported by Demin et al. Mol Cell, 2021, a study cited in the original version but not previously discussed in depth. In addition, we now

cite Adamowicz et al. *Nat Cell Biol.* 2021 in the context of transcriptional recovery after DNA damage.

In the Introduction after the sentence “Loss of XRCC1 has been shown to cause elevated poly-ADPr levels through PARP1 hyperactivation, which has been linked to neurodegeneration^{8,9}.” we have added “By restraining PARP1 recruitment and catalytic activity during base excision repair (BER), XRCC1 prevents persistent PARP1 binding to BER intermediates. Recent work has further clarified how XRCC1 loss influences the temporal dynamics of PARP1 signaling. In particular, Demin et al.¹⁰ demonstrated that in MMS-treated XRCC1-deficient cells, early PARP1 hyperactivation drives rapid NAD⁺ consumption, which in turn suppresses PARP1 auto-modification at later time points. These findings indicate that XRCC1 not only restrains PARP1 activation but also preserves NAD⁺ availability necessary for sustained ADPr responses. Moreover, sustained PARP1 activity induced by XRCC1 deficiency has been shown to inhibit transcriptional recovery after DNA damage¹².”

In the Results section, we have revised the sentences to read as follows:

“We observed higher mono- and poly-ADPr levels in WT compared to XRCC1 KO cells after recovery from methyl methanesulfonate (MMS) treatment (Supplementary Fig. 5C). This late decrease in PARP1 activity in XRCC1-deficient cells is consistent with the model in which early PARP1 hyperactivation accelerates NAD⁺ consumption, thereby limiting auto-modification and polymer extension at later time points¹⁰.”

“However, after 90 minutes of MMS-induced DNA damage, WT cells showed higher levels of poly-ADPr and core histone mono-ADPr than XRCC1 KO cells (Supplementary Fig. 5E), in line with reduced PARP1 activity at later stages of damage in XRCC1-deficient cells⁸.” Regarding H1 mono-ADPr we have now added: “This identification was further confirmed by mutating the main ADPr sites on H1^{15,42} (Fig. 5E). The sustained mono-ADPr signal on H1 in XRCC1-deficient cells is intriguing, as studies indicate that mono-ADPr weakens H1–DNA interactions and can promote its displacement from chromatin during DNA damage⁵², a process that may contribute to chromatin relaxation at damage sites. Prolonged H1 mono-ADPr may therefore reflect extended chromatin remodeling resulting from persistent repair intermediates.” At the end of the section on ADP-ribosyl-ubiquitylation just before the Discussion we have added the conclusion: “This highlights the influence of XRCC1 on downstream ADPr and ubiquitin signaling during the DNA damage response and could imply a direct link between XRCC1 and the various effects suggested for ADPrUb substrates including their degradation, stabilization or relocalization^{35,36,53}.”

We have expanded the Discussion to integrate our findings with the model proposed by Demin et al. Mol Cell, 2021: “As demonstrated by Demin et al.¹⁰, XRCC1 prevents excessive and prolonged PARP1 engagement with repair intermediates during BER, thereby acting as a PARP1 anti-trapper that ensures efficient, unobstructed DNA repair. In the absence of XRCC1, PARP1 becomes persistently associated with BER intermediates, leading to an early burst of PARP1 catalytic activity. This initial hyperactivation depletes NAD⁺, thereby limiting PARP1-mediated poly-ADPr at later time points¹⁰. Importantly, the ADPr dynamics in XRCC1-deficient cells differ markedly between MMS and H₂O₂, reflecting how XRCC1 loss interacts with the distinct repair pathways engaged by these agents. MMS generates alkylated bases that produce a high burden of BER intermediates, which accumulate in the absence of XRCC1 and promote early PARP1 trapping and hyperactivation¹⁰, whereas H₂O₂ induces strand breaks and oxidative lesions that enter BER through different entry points and with distinct processing kinetics. Although the precise mechanisms remain unresolved, these XRCC1-dependent differences in BER intermediate load and repair dynamics are consistent with the distinct ADPr responses we observe for MMS and H₂O₂. To our knowledge, no prior study has directly compared mono- and poly-ADPr dynamics across these two damage types in XRCC1-deficient cells, making this stimulus-dependent divergence a novel observation enabled by the sensitivity of our antibody toolkit.”

2. Related to the above point, an important experiment would be to determine the impact of PARG inhibition on the formation of excess mono-ADPr in XRCC1 KO cells. This would allow a distinction between a possible role of XRCC1 in influencing direct PARP1-dependent mono-ADP-ribosylation vs XRCC1 affecting the formation of mono-ADPr as a result of a potential impact on PARG-dependent poly-ADPr hydrolysis.

We appreciate this insightful suggestion. We agree that examining the effect of PARG inhibition would help discriminate between direct PARP1-dependent mono-ADP-ribosylation and secondary formation due to poly-ADPr turnover.

PARG inhibition experiments have proven more challenging than originally expected, due to the presence of PARGi-induced mono-ADPr signals that become apparent only

when serine-linked mono-ADPr levels are relatively low. This likely reflects the induction of mono-ADPr species other than serine mono-ADPr, which is not hydrolyzed by PARG. In fact, recent studies have shown that mono-ADPr on tyrosine, as well as on aspartate and glutamate, is cleaved by PARG (Longarini & Matic, *Nat. Commun.* 2024; Rack et al., *JBC* 2024) and is therefore increased upon PARG inhibition.

Therefore, the scope of the suggested PARGi experiment is somewhat limited, with feasibility restricted to those conditions and time points at which we observed the highest mono-ADPr levels induced by DNA damage. As shown in the below figure, conditions with lower serine mono-ADPr levels present analytical challenges due to the presence of mono-ADPr signal induced by PARG inhibition, making it difficult to specifically assess the effect of PARG inhibition on serine mono-ADPr. On the other hand – and these are conditions we selected for our experiments – this problem disappears at certain time points of continuous H₂O₂ treatment, where serine-linked mono-ADPr reaches sufficiently high levels to allow unambiguous analysis. Building on the short, late PARG inhibition approach of Demin et al. (*Mol Cell* 2021), we applied multiple PARGi pulses (5, 10, and 15 min before harvest following 25 min of H₂O₂ treatment) to directly analyse the turnover dynamics of ADPr.

As shown in the new Figure 6, which directly addresses the reviewer's suggestion to examine PARG-dependent mono-ADPr turnover, and as detailed in the revised text below, our results provide no evidence that XRCC1 deficiency affects PARG activity. By degrading poly-ADPr, PARG effectively functions as a poly-to-mono conversion enzyme. However, as outlined above, our new data do not support a role for XRCC1 in regulating PARG activity.

We have dedicated a new chapter to the new Figure 7 that reads as follows:

“Altered mono-ADPr dynamics in XRCC1-deficient cells involve PARG-dependent turnover

Having observed that mono-ADPr levels increase in the absence of XRCC1 (Figs. 5, 6) we next investigated whether this signal arises from altered PARG-dependent turnover of poly-ADPr. To address this, we examined the effect of PARG inhibition (PARGi) on the accumulation of serine mono-ADPr in XRCC1 KO cells. Although PARG can remove mono-ADPr from tyrosine and aspartate/glutamate^{17,44}, it cannot hydrolyze

serine-linked mono-ADPr²¹. PARG inhibition therefore provides a means to assess how much serine mono-ADPr arises from poly-ADPr degradation. Consistent with our previous studies^{18,21}, when PARG inhibition is applied throughout genotoxic treatment, mono-ADPr is largely PARG-dependent, as evidenced by the strong reduction in mono-ADPr signal (Fig. 7A). Notably, the extent of this reduction is similar in WT and XRCC1 KO cells, consistent with elevated mono-ADPr levels in XRCC1 KO cells arising from increased levels of the PARG substrate poly-ADPr rather than from increased PARG activity.

Next, we extended the late-stage PARG inhibition approach of Demin et al. by applying short PARGi pulses (5, 10, or 15 min before harvest) during continuous 45 min H₂O₂ treatment, the condition showing the largest XRCC1-dependent increase in mono-ADPr relative to WT cells. This strategy provides a direct readout of acute PARG-dependent conversion of poly-ADPr to mono-ADPr, allowing us to determine whether persistent mono-ADPr observed at late time points in XRCC1 KO cells is generated by rapid hydrolysis of pre-formed poly-ADPr. Consistent with low poly-ADPr levels at this late stage of ADPr signalling, short pulses of PARG inhibition had only a mild effect on mono-ADPr (Fig 7B).

Together, our results provide no evidence that XRCC1 deficiency measurably alters PARG activity. Instead, they indicate that mono-ADPr accumulation in XRCC1-deficient cells reflects a time-dependent contribution of PARG-mediated turnover, with strong dependence on PARG activity at early stages revealed by full PARG inhibition (Fig. 7A), but only limited sensitivity to late-stage PARG inhibition (Fig. 7B).”

Minor comments:

3. Sup. Fig 2 would be a little easier to interpret if the diagnostic ions in panels A and B were exactly the same and displayed in the same order. Also, it would be helpful to add an extra column in table Sup. Fig 2B indicating which ions are present for both modifications, and which are specific for poly-ADPr.

We thank the reviewer for this suggestion and we agree that the interpretation of the diagnostic ions could be clarified. We have now aligned the diagnostic ions shown in Panel A with those listed in the table in Panel B, ensuring they appear in the same order. In addition, the rows representing poly-ADPr-specific diagnostic ions in Panel B are now highlighted in grey, which we believe provides a clearer visual explanation than adding an additional column. The corresponding figure legend for Supplementary

Figure 2 has been updated with the following sentence: 'Poly-ADPr-specific ions are highlighted in grey.'

4. The fairly frequent formation of adenosine (both from mono and poly-ADPr) indicates that cleavage at the glycosidic bond is frequent during MS fragmentation. Have the authors considered looking for other ions in which the adenosine moiety is missing?

Thank you for this comment — we agree that this point required clearer explanation in the manuscript. With respect to diagnostic peaks, the terminal (last-added) ADP-ribose of a poly-ADPr chain is indistinguishable from mono-ADPr: both can generate the same major diagnostic fragment ions (adenine, adenosine, AMP, ADP). This means — and this is the clarification we have now added — that there are no ions produced by mono-ADPr that are absent from poly-ADPr. In contrast, the new diagnostic ions introduced in our manuscript are specific for poly-ADPr. As a consequence, in poly-ADPr it is not possible to determine whether an observed adenosine fragment originates from the terminal ADP-ribose unit or from cleavage of an internal glycosidic bond. For the same reason, such spectra do not allow conclusions about the frequency of glycosidic bond cleavage.

Regarding the reviewer's question related to the comment, we did search for ions lacking the adenosine moiety, as suggested. However, none were detected. This is expected, as we analyze only positively charged ions: adenine-containing fragments are readily detected in positive mode, whereas ions lacking adenine (e.g., ribose or phospho-ribose fragments) are neutral or negatively charged and therefore cannot be observed under these conditions.

To clarify in the manuscript that mono-ADPr does not generate any diagnostic ions that are absent from poly-ADPr, we have added the following text after the sentence "From the structure of di-ADP-ribose, we interpreted these ions as corresponding to singly charged AMP+phosphoribose, ADP+phosphoribose, ADPr+AMP and di-ADPr (Fig. 2A and Supplementary Fig. 2A,B)": "These additional ions are specific for poly-ADPr because their formation requires two covalently attached ADP-ribose moieties. In contrast, mono-ADPr diagnostic ions can originate either from the terminal ADP-ribose within a poly-ADPr chain or from mono-ADPr."

5. The resolution of some of the MS spectra in the files I received was insufficient to properly read the y- and b-series fragment numbering on the peptide backbones at the top of the spectra (particularly Fig. 2), or to clearly distinguish between (colored) fragment ion peak heights and the lines (are they dotted?) used to label them.

Thank you for pointing this out — the text and peak numbering in the peptide backbone scheme were indeed too small. We have now increased the scheme and font size to match that used in our recent *Nature Chemical Biology* paper (Kolvenbach, Palumbieri et al., 2025). We also increased the gap sizes of the dotted lines to the respective peak labels, to clarify the separation of peaks and lines. In addition, we ensured that the revised manuscript contains spectra at sufficiently high resolution to ensure readability.

6. Many of the western blot panels are cropped to a limited number of samples and then displayed side-by-side for comparison. This is of course necessary when different ADPr detection reagents are used (Fig. 3, 4 and Sup. Fig.4), but hampers adequate comparison between samples during a time course (Fig. 5B and Sup. Fig 5D, E).

We thank the reviewer for this comment and fully understand the concern. The time-course blots in Figs. 5B and Sup. 5D,E were separated because identical exposure settings cannot be used across all time points, as shown in the figure below. The earlier time points contain substantially stronger ADPr signals, which would completely obscure the weaker later time points if the same exposure were applied. For this reason, the early and late time points required different exposure settings, so they are shown as separate panels. This approach avoids overexposure artifacts and enables accurate visualization of both weak and strong signals. To make this clearer, we now state this explicitly in the figure legends and have added the corresponding exposure times to each panel. Importantly, the full sequential comparison across all time points is shown in Fig. 6, where the signal intensities fall within the same dynamic range and consistent exposure can be used.

7. It is unclear what the “nuclear intensity” values used to plot microscopy quantifications represent. How were pixel intensities normalized to the arbitrary units in the ca. 1-100 range? Was background subtraction performed? Also, it seems that mean \pm SEM were determined as if each nucleus was an independent replicate (n is as high as 80 in some panels). This is incorrect. Instead, the values for all quantified nuclei for each biological replicate should be averaged, so that a mean intensity value for each biological replicate is obtained. These mean values per replicate are then used to derive a mean \pm SEM for the experiment, with $n=3$.

We thank the reviewer for pointing this out. In the original submission, the microscopy quantifications were indeed taken from a single representative experiment and plotted as mean gray values of individual nuclei. As the reviewer correctly notes, this approach treats each nucleus as an independent measurement and is therefore not appropriate

for deriving mean \pm SEM. As stated in the cited paper in Material and Methods, we performed background subtraction on all measured values.

We have now replotted all microscopy data according to the reviewer's recommendation. Specifically:

1. For each experiment, the mean nuclear intensity was calculated per biological replicate, so that individual nuclei are not treated as independent data points.
2. These per-replicate means (n = 3 biological replicates) were used to compute the overall mean \pm SEM now shown in the updated figures.
3. To facilitate comparison across different conditions and between antibodies, we have normalized all values to the signal of untreated WT cells.

All figures, quantification plots, and legends have been updated accordingly to reflect this corrected analysis and normalization procedure.

Typos:

- Fig 1A, top panel – “Stragety” should be “strategy”

- Fig. 2A, zoomed-in area of tri-ADPr peptide - ion 889.1315 label should be “AMP+ADPr”

- Line 157 – “AMP+ADP” cannot be correct. Maybe “ADPr+ADP”?

We thank the Reviewer for pointing out these typos, which we have now corrected. Regarding the last typo, we have also revised the sentence from "From the structure of di-ADP-ribose, we interpreted these ions as corresponding to singly charged ADP+AMP, ADPr+AMP, AMP+phosphoribose, ADP+phosphoribose (Fig. 2A and Supplementary Fig. 2A,B)." to "From the structure of di-ADP-ribose, we interpreted these ions as corresponding to singly charged AMP+phosphoribose, ADP+phosphoribose, ADPr+AMP and di-ADPr (Fig. 2A and Supplementary Fig. 2A,B)."

Reviewer #2 (Remarks to the Author):

This work reports the development of synthetic antibodies specific for poly(ADP-ribose) using chemoenzymatically prepared peptides with varied numbers of ADP-ribose and affinity maturation of synthetic antibodies specific for mono-ADP-ribose. Then, authors used selected antibodies reagents for poly(ADP-ribose) and mono-ADP-ribose to study ADP-ribosylation derived from PARP1 in XRCC1-deficient cells. Due to technical difficulties in preparing antigenic peptides carrying poly(ADP-ribose), authors employed a different blocking strategy by utilizing peptides with mono-ADP-ribose. This way allowed identification of new antibody reagents with improved sensitivity. Through phage display-aided selection, an antibody recognizing mono-ADP-ribosylated peptides in high specificity was discovered, showing different recognition patterns from a previously published antibody for mono-ADP-ribosylation. Combining the developed antibodies for poly(ADP-ribose) and mono-ADP-ribose, including ones reported in this manuscript and one from a previous study (abd43647), authors revealed time-dependent mono- and poly-ADP-ribosylation in wild-type and XRCC1 KO cells, and an increase of ADP-ribosyl-ubiquitylation in XRCC1 KO cells upon DNA damage. The significant efforts by authors in developing and affinity maturing antibody reagents for different forms of ADP-ribosylation are commendable. But the overall technical advances in generating ADP-ribosylation-specific reagents seem incremental compared with previously published studies. While the antibody specific for poly(ADP-ribose) showed higher sensitivity, new knowledge or information revealed exclusively by this reagent is little. Likewise, a new antibody more specific for mono-ADP-ribosylation was discovered from affinity maturation, but a previously published antibody with comparable affinity and specificity could possibly address the same questions. In addition to the technical novelty and application of the reported antibodies, the findings from XRCC1 KO cells about the ADP-ribosylation dynamics and the ADP-ribosyl-ubiquitylation provide limited insights into the functions and molecular mechanisms of mono- and poly-ADP-ribosylation and composite modification related to XRCC1. Only ADP-ribosylation dynamics under different DNA damaging conditions and increased ADP-ribosyl-ubiquitylation were shown. Functional and mechanistic understandings from these observations would be critical to support the essential roles of the reported toolbox in studying XRCC1-deficiency driven ADP-ribosylation. Given the modest technical innovation, already published antibody reagents with comparable affinity and specificity, and lack of functional and mechanistic insights from the XRCC1 deficiency-related findings, this manuscript is not recommended for publication in Nature Communications.

We would like to thank Reviewer #2 for the detailed summary of the main points of our manuscript, although we note that their overall evaluation was more negative than that of the other reviewers. By contrast, the other two reviewers both emphasized the importance, high quality and relevance of our study, with Reviewer #1 describing it as “of current interest in the ADP-ribosylation field” and noting that the “overall quality of the data is excellent”, and Reviewer #3 characterizing it as “an important contribution to the protein ADP-ribosylation field” and “a rigorous and interesting study”.

We are pleased that the reviewer raised no concerns regarding the technical quality or robustness of the experimental work and are also grateful for the positive acknowledgement that “The significant efforts by authors in developing and affinity maturing antibody reagents for different forms of ADP-ribosylation are commendable.” We would like to emphasize that these efforts, together with our novel phage-display blocking strategy, have yielded antibodies that provide clear and demonstrable advantages over existing tools. This assessment is fully supported by the other reviewers, who raised no concerns regarding the utility or novelty of our reagents and instead explicitly recognized their value, including their specificity and sensitivity. Reviewer #1 highlighted that “the new reagents and methodologies described here will certainly influence further studies in the field” and Reviewer #3 noted that “The new anti-poly antibodies do appear particularly sensitive and specific.”

Already in the original version of the manuscript, we illustrated the distinct binding preferences of the new mono-ADPr antibody: “AbD41122 did not recognize free mono-ADP-ribose detected by AbD43647. Its binding to the different peptides showed context dependencies similar to those of its parental clone, but distinct from AbD43647 (Fig. 4B), illustrating the complementarity of the two mono-ADPr antibodies”. Nevertheless, we also recognize that we may not have sufficiently highlighted the comparative advantages of our antibodies over existing tools, even though these data are already included in the manuscript. In a long and technically detailed study such as ours, these aspects may not always be immediately apparent on first reading. To address this and to further strengthen our case, we have now added additional text to better explain and emphasizes the advantage of our new reagents.

“Importantly, our analyses highlight that mono-ADPr antibodies differ substantially in substrate preference, and no single clone is sufficient to capture the full biological diversity of mono-ADPr (Supplementary Fig. 4A). Moreover, AbD41122 robustly

detects starvation-induced mono-ADPr on RNA⁴⁹, while AbD43647 does so only weakly and with nonspecific background (Fig. 4C). These differences reflect distinct biological recognition profiles, underscoring the necessity of having multiple high-quality mono-ADPr antibodies rather than relying on a single reagent. Thus, our expanded mono-ADPr antibody panel provides coverage of the diverse biochemical contexts in which mono-ADPr occurs.”

Importantly, as detailed below, we have also performed additional experiments that illustrate specific application areas in which our antibodies enable analyses that are not achievable with existing tools.

First, we showed that the exceptional sensitivity of our new antibodies enables detection of extremely low cellular poly-ADPr levels – specifically endogenous poly-ADPr in RPE1 cells in the absence of DNA damage treatment – that remain entirely undetectable with existing tools. This is important because the breakthrough discovery by the Caldecott lab that PARP1 senses unligated Okazaki fragments has stimulated growing interest in basal ADPr levels. Our new tools enable such studies without the need to artificially elevate poly-ADPr by inhibiting PARG, or to restrict analyses to cell lines with high basal poly-ADPr levels, such as U2OS. The text describing this new experiment reads as follows:

“This high sensitivity is particularly critical in cellular contexts with very low basal ADPr levels, such as RPE1 cells grown under hypoxia, (Fig. 3E). Even under these challenging conditions, our poly-ADPr antibodies detected endogenous PARP1-dependent poly-ADPr, whereas commercial reagents failed to produce a detectable signal.”

Second, to further illustrate the advantages arising from the distinct binding preferences of our new affinity-matured mono-ADPr antibody AbD41122, compared with AbD43647—our previously introduced mono-ADPr antibody (Longarini et al., Mol Cell 2023) and widely regarded as the tool of choice for specific mono-ADPr detection—we drew on data from an ongoing collaboration with an RNA-focused laboratory that is applying the new tools. The new data (new Fig. 4C), which were initially intended for a separate publication, have been incorporated into this manuscript to address the reviewer’s broader concerns and clearly show that only the new antibody AbD41122 robustly detects RNA mono-ADPr. In the revised manuscript, this experiment is described as follows: “Moreover, AbD41122 robustly detects starvation-induced mono-ADPr on RNA⁴⁹, while AbD43647 does so only weakly and with nonspecific background (Fig. 4C).”

Regarding Reviewer's comment related to the functional and mechanistic insights, we would like to clarify that the manuscript is most appropriately viewed in the Resource/Technology format. This could not be explicitly indicated during submission, as *Nature Communications* publishes all papers under a single "Article" designation. In this context—and similar to our Bonfiglio et al., *Cell* 2020 study, which was published in the Resource format—Figures 5–7 are intended primarily as proof-of-principle demonstrations, illustrating how the new tools and methodologies will enable future investigation of key questions across diverse biological processes. This is a defining feature of the Resource/Technology format, which places emphasis on tool development and validation rather than on extensive functional or mechanistic analyses. Nevertheless, as detailed above in our reply to Reviewer #1, we have added a new set of experiments (new Fig. 7) analyzing the impact of PARG inhibition on the formation of excess mono-ADPr in XRCC1-deficient cells.

Reviewer #3 (Remarks to the Author):

The new study by Dauben et al. makes an important contribution to the protein ADP-ribosylation field, particularly in terms of providing new tools for studying this modification.

Regarding the tools, the manuscript primarily describes a strategy for selecting sensitive poly(ADP-ribose) antibodies; the development and characterisation of these antibodies for both immunoblotting and cell immunostaining applications; and the affinity-maturation of some previously developed site-specific and general mono(ADP-ribose) antibodies. The new anti-poly antibodies do appear particularly sensitive and specific.

In terms of biological insight, the authors apply some of these tools to show that cellular XRCC1 deficiency results in an increase in different ADP-ribosylation types including poly(ADP-ribosyl)ation, mono(ADP-ribosyl)ation, and a composite ubiquitin-mono(ADP-ribosyl) modification, the last mentioned possibly accounting for the increased accumulation of RNF114 (the reader of the composite modification in question) at the sites of laser-induced damage.

The study also contains further interesting observations, such as the somewhat different spatial distribution of mono- vs poly(ADP-ribose) signals in cells (in Suppl. Fig. 3D - by the way, can the extent to which these signals are “punctate” or not be quantified?) or - this is particularly interesting - the identification of a signature di(ADP-ribose) MS fragment that is specific for poly(ADP-ribosylation) and seems to have confounded previous site identification efforts by mimicking a double mono(ADP-ribose).

It is a rigorous and interesting study and I don't have any major points of criticism. Here are some minor points that could be taken into account to further improve it:

We thank the reviewer for their very positive assessment of our work and for describing it as rigorous, interesting and an important contribution to the ADP-ribosylation field. We are pleased that the reviewer did not raise any major concerns following their thorough and detailed evaluation of our manuscript. As detailed below, we have addressed all of the minor points, which we appreciate as constructive and helpful in improving the manuscript. In particular, we have provided a substantial set of new experiments to validate the specificity of our poly-ADPr antibodies from multiple angles (point #1), including the development of a simple yet powerful method for separating mono-, di-, tri-, tetra-, and penta-ADPr peptides.

Regarding the question about the “punctate” signals, although we do observe punctate poly-ADPr signals in our immunofluorescence images, these structures differ substantially from classical DNA-damage foci such as γ -H2AX or RAD51. Instead of a small number of well-separated, sharply defined foci, poly-ADPr staining produces numerous very small, variably sized puncta per nucleus, many of which partially overlap or lie close to background fluctuations. Because of this high density and the absence of clear, contiguous boundaries, standard segmentation-based quantification (as used for γ -H2AX or RAD51) would require arbitrary thresholding and would therefore not yield robust or interpretable measurements.

We agree, however, that the distinct spatial distribution is biologically interesting, as this pattern could reflect localized activation of PARP enzymes at specific chromatin sites, where polymeric ADP-ribose accumulates and recruits polyADPr-binding proteins. In contrast, mono-ADPr shows a more diffuse nuclear distribution, consistent with its broader roles across chromatin. Reliable quantification of the punctate poly-ADPr pattern would require higher-resolution imaging and dedicated analysis pipelines capable of resolving individual submicron puncta, which would be valuable to pursue in future dedicated studies.

To describe this point more clearly in the manuscript, we have revised the description of Supplementary Fig. 3G to read: “Poly-ADPr staining appeared as nuclear foci. This pattern likely reflects the localized activation of PARP enzymes at specific chromatin sites, where polymeric ADP-ribose accumulates and recruits polyADPr-binding proteins. In contrast, mono-ADPr exhibited a more diffuse nuclear distribution, consistent with its broader roles across chromatin.”

1) Are the new poly(ADP-ribose) antibodies specific for Ser-attached poly(ADP-ribose), or are they attachment-independent? Would they recognise Glu/Asp-linked poly(ADP-ribose)? Would they recognise free poly(ADP-ribose)? Could the authors discuss insights they might already have at their disposal that suggest answers to these questions? Would it take a lot of effort to explore this question further experimentally? I think it would be useful to at least discuss it. Similarly, do the authors think these antibodies might unequally detect chains differing in length or branching status?

We thank the reviewer for these insightful questions regarding the specificity of our poly-ADPr antibodies. To address these points, we have conducted an extensive series of additional experiments designed to further characterize their specificity and binding properties. First, our poly-ADPr antibodies are not restricted to serine-linked ADPr, as they robustly detect PARP1-derived polymer generated both in the presence and absence of HPF1, indicating recognition that does not depend on the underlying amino acid linkage and is therefore compatible with Glu/Asp-linked and other forms of poly-ADPr (new Fig. 3C). They also detect free poly-ADP-ribose with high sensitivity, demonstrating that a protein or peptide context is not required for binding (new Supplementary Fig. 3C). Most notably, after testing various conditions, we established a gel-based approach that separates the differently modified peptide—used as antigen for antibody generation—into their individual components: mono-, di-, tri-, tetra- and penta-ADPr. This simple and easily implementable method revealed that the new antibodies can recognize poly-ADPr chains as short as di-ADPr (new Supplementary Fig. 3B).

Taken together, our new data demonstrate that the poly-ADPr antibodies exhibit broad attachment independence and robust detection across multiple polymer lengths, including short-chain and free polymer. Assessing antibody recognition of branched poly-ADPr is technically not feasible, because it is not possible to generate pure branched polymer without co-purifying linear species, making it impossible to

determine whether the antibodies specifically recognise branch points. In practical terms, however, the antibodies will detect branched poly-ADPr, since branching is relatively infrequent—typically occurring in only ~1–3% of ADPr units (i.e., roughly one branch every 20–50 residues)—meaning that 97–99% of the polymer is linear and therefore readily recognised by the antibodies.

To describe these new experiments we have added the following text: “

Beyond their high sensitivity, our antibodies are not restricted to serine-ADPr but also detect poly-ADPr on aspartate and glutamate, indicating that they recognise poly-ADPr irrespective of the underlying amino acid (Fig. 3C). By establishing an optimized electrophoretic separation method for resolving oligo-ADP-ribosylated peptides, we found that the antibodies recognize poly-ADPr chains as short as di-ADPr (Supplementary Fig. 3B). Moreover, they efficiently detect free poly-ADP-ribose (Supplementary Fig. 3C). Together, these results illustrate the broad substrate recognition of the new poly-ADPr antibodies.”

2) In terms of citations, the selection of references is generally relevant and complete, but the authors could cite the 1960s studies they mention in line 39. The statement “Dysregulation of ADPr is implicated in various diseases, including cancer and neuro-degeneration, ...” in lines 40-41 could also do with a citation supporting these claims. For the composite Ub-ADP-ribose modification in line 94, the Zhu et al. Sci Adv paper could be referred to alongside the cited cellular studies.

We thank the reviewer for noting that our selection of references is appropriate and complete. We have followed the reviewer’s advice and added the missing references. For the initial identification of ADPr in the 1960s, we have included the study that first described the formation of poly-ADPr. Regarding the implication of ADPr in disease, we have now cited Curtin, N.J. & Szabo, C. Poly(ADP-ribose) polymerase inhibition: past, present and future. Nat Rev Drug Discov 19, 711–736 (2020). In addition, in line 98 (previously 94), we have added a reference to Zhu et al., Sci Adv, which was already cited elsewhere in the manuscript.

3) The sentence in lines 55-58 (“A prime example...”) is a bit convoluted; the authors could consider simplifying it and perhaps splitting it in two.

We agree with the reviewer that this sentence is convoluted. Given that an already lengthy manuscript has become even longer with the revision, we have simplified and shortened it to: “A prime example of this is serine ADPr, discovered less than ten years ago despite its abundance”

4) I appreciate the fact that immunoblots are generally accompanied by Ponceau S stain images serving as loading controls, but these are sometimes a bit dark and low in contrast, especially when printed out - perhaps worth checking and improving their clarity.

We thank the reviewer for pointing this out. Indeed, some of the Ponceau-stained blots appeared too dark. We have adjusted the brightness and contrast to ensure that all Ponceau images are equally clear and consistent in presentation. We note that in a few cases the contrast of the Ponceau staining remains limited. This is a consequence of intentionally loading low amounts of total protein to ensure that the highly sensitive ADPr antibody signals remain within a non-saturated detection range. While we tried to optimize Ponceau staining for all blots, in some cases the reduced protein load inherently limited the achievable Ponceau signal intensity.

5) In lines 283-285, a background signal that can be eliminated with sulphuric acid is mentioned. Could the authors comment on what this signal might be and why it gets eliminated with sulphuric acid?

Compared to total cell lysis, lysis with sulfuric acid enriches for histones, which remain soluble due to their high positive charge, while most other proteins precipitate. Thus, the reduction in background signal is not a direct effect of sulfuric acid itself but rather a consequence of histone enrichment. We are grateful for this comment, as it made us realize that our original wording was unclear. We have therefore revised the sentence to read: “...a general background signal, which was largely eliminated when histones were enriched by sulfuric acid extraction during cell lysis” (now lines 321-323)

6) Could the authors explain in clearer words the term “antibody leakage” used in line 300?

We thank the reviewer for pointing this out. By “antibody leakage,” we refer to the phenomenon in which antibody molecules used for pulldown detach from the beads and appear as background bands on immunoblots, thereby complicating data interpretation. We have clarified this in the revised manuscript, which now reads: “...antibody leakage, i.e. antibody molecules released from pulldown beads that appear as background bands on immunoblots.”

7) In line 332, could the authors clarify that they used both Olaparib and the more PARP1-specific AZD9574 to establish this conclusion?

To clarify this, we have added the following to the sentence: “...using either Olaparib or the PARP1-specific inhibitor AZD9574...”

8) The procedure involving perchloric acid enrichment and mutation used to identify H1 as one of the bands could be briefly explained in lines 337-338. This fragment contrasts with otherwise more thorough explanations of the approaches applied.

We have followed the reviewer’s suggestion and briefly explained the procedure. The new text reads: “Through perchloric acid enrichment, which solubilizes histone H1 while leaving core histones and other proteins insoluble (Supplementary Fig. 5G), we identified this mono-ADPr–exclusive target as histone H1 (Fig. 5D). This identification was further confirmed by mutating the main ADPr sites on H1^{14,41} (Fig. 5E).”

9) Lines 357-359: Is the more spatially heterogeneous distribution of poly(ADP-ribose) signal across cells due to the signal having already disappeared in some cells, or was it not there in some of them in the first place? Is it possible to distinguish between the two? What do the authors consider more likely?

We thank the reviewer for this thoughtful question. We consider the heterogeneous distribution of poly-ADPr to most likely reflect its highly transient nature, as poly-ADPr is rapidly synthesized and degraded. Consequently, small differences in the timing or

magnitude of PARP1 activity between individual cells can lead to pronounced variability in detectable poly-ADPr signal. In contrast, mono-ADPr is more persistent and therefore less affected by such temporal or cell-to-cell fluctuations.

We have clarified this in the manuscript by adding the following sentence: “This more heterogeneous distribution of poly-ADPr likely reflects its transient nature, whereby rapid turnover renders its detection highly sensitive to subtle temporal or cell-to-cell differences in PARP1 activity.”

10) The sentence “This raises the question whether the toxic accumulation of ADP-ribosyl-linked serine ubiquitylation contributes...” sounds as if it was known that ADP-ribosyl-linked Ub was toxic, and the remaining question was whether it contributed or not to these particular symptoms. However, the accumulation of this composite modification should only be called toxic if it does cause these symptoms and this is an open question as the authors admit. In short, perhaps it’s better not to call it “toxic” until proven.

We completely agree with the reviewer that the term “toxic” should not be used here, as this remains an open question. We have therefore removed “toxic” from the sentence.

11) Line 552: space in E. coli missing. Also, some spaces between numbers and units are missing in Methods.

We thank the reviewer for noticing these typographical errors. We have corrected the missing space in *E. coli* and ensured consistent spacing between numbers and units in the Methods section.

13) In Fig. 1A, it looks a bit as if the circle representing ADP-ribosyl was attached next to a serine (“S”) rather than to a serine. Maybe it is just me

We thank the reviewer for noticing this detail. We have adjusted the schematic in Fig. 1A so that the circle representing ADP-ribose is now centered on the “S,” more accurately depicting attachment to the serine residue.

14) Overall, the figures are high quality and clear and the legends are exhaustive, providing information on replicability, number of cells etc. The Methods description seems exhaustive.

We thank the reviewer for this positive assessment and are pleased that they found the figures clear, of high quality, and accompanied by sufficiently detailed legends and Methods descriptions.

15) The authors could consider changing poly-ADP-ribosylation etc. to poly(ADP-ribosylation) - but I guess it is their conscious choice to use the hyphenated form, which I respect.

We have followed the reviewer's suggestion and adopted the poly(ADP-ribosylation) notation in the Abstract. In the main text, we have retained the shorter hyphenated form poly-ADPr for conciseness and readability, as well as for consistency with the term mono-ADPr.

Monday, February 16, 2026

Point-by-point answer to Reviewers:

NCOMMS-25-57878A (Versatile and sensitive detection of mono- and poly(ADP-ribose)ation reveals XRCC1-dependent remodeling of PARP1 signaling)

Summary:

We would like to thank the editor and the reviewers for their time and careful evaluation of our revised manuscript. We were encouraged by the indication that most points raised in the initial review had been satisfactorily addressed and the positive assessments of Reviewers #1 and #3, who now consider the manuscript suitable for publication. Below, we provide a detailed, point-by-point response to all comments on the revised manuscript. Several points relate to data already included in the original submission, while others concern a newly added experiments introduced during the revision. In all cases, we have carefully addressed the points raised and further strengthened the manuscript with substantial additional experiments.

- We performed sets of additional experiments further comparing AbD41122 against the commercial reagents suggested by Reviewer #2 and included quantitative analysis of immunoblots
- We further clarified the interpretation of sensitivity versus affinity in Fig. 3B and expanded the methodological explanation of the SpyTag modular system.
- We revised the wording regarding olaparib to clearly distinguish it as a PARP inhibitor from the PARP1-specific inhibitor AZD9574.
- Minor figure labeling issues identified by the reviewers have been corrected.
- In response to Reviewer #3's suggestion, we revised the title to better emphasize the methodological advances of the study.
- The manuscript has been shortened with changes highlighted in yellow. Further reduction of the text was not feasible without compromising clarity and completeness, given the number of experiments presented and the substantial additional data and explanations requested during the two rounds of revision.

We believe we have significantly improved the manuscript by following reviewers' suggestions, incorporating new results and clarifications, and addressing their concerns. We hope the revised version is now suitable for publication in *Nature Communications*.

Previous version	Revised Version	
	Fig. 1 Fig. 2 Fig. 3 Fig. 4 Fig.5 Fig. 6 Fig. 7 Fig 8.	
-	Fig. 2 Fig. 2a Fig. 2b Fig. 2c Fig. 2d	
-	Fig. 3 Fig. 3a Fig. 3b Fig. 3c Fig. 3d Fig. 3e Fig. 3f	
-	Fig. 4 Fig. 4a Fig. 4b Fig. 4c Fig. 4d Fig. 4e Fig. 4f Fig. 4g	Revised
-	Fig. 5 Fig. 5a Fig. 5b Fig. 5c Fig. 5d	
-	Fig. 6 Fig. 6a Fig. 6b Fig. 6c Fig. 6d	
-	Fig. 7 Fig. 7a Fig. 7b	
-	Fig. 8 Fig. 8a Fig. 8b	

New	SupplementaryFig.1 SupplementaryFig.2 SupplementaryFig.3 SupplementaryFig.4 SupplementaryFig.5 SupplementaryFig.6 SupplementaryFig.7	
-	SupplementaryFig.1 SupplementaryFig.1a SupplementaryFig.1b SupplementaryFig.1c SupplementaryFig.1d	
-	SupplementaryFig.2 SupplementaryFig.2a SupplementaryFig.2b SupplementaryFig.2c	Revised
-	SupplementaryFig.3 SupplementaryFig.3a SupplementaryFig.3b SupplementaryFig.3c SupplementaryFig.3d	New Revised New New
Supl 3a	SupplementaryFig.3e SupplementaryFig.3f SupplementaryFig.3g SupplementaryFig.3h	New New
Supl 3b		
Supl. 3c		
-	SupplementaryFig.4 SupplementaryFig.4a SupplementaryFig.4b SupplementaryFig.4c SupplementaryFig.4d	
Supl 3d		
Supl 3e		
Supl 3f		
Supl 3g		
-	SupplementaryFig.5 SupplementaryFig.5a SupplementaryFig.5b SupplementaryFig.5c SupplementaryFig.5d SupplementaryFig.5e SupplementaryFig.5f SupplementaryFig.5g	New New
Supl 4a	SupplementaryFig.5h	
Supl 4b	SupplementaryFig.5i	
Supl 4c	SupplementaryFig.5j	
Supl 4d	SupplementaryFig.5k	
Supl 4e	SupplementaryFig.5l	
Supl 4f		
Supl 4g		
Supl 4h		
Supl 4i		
Supl 4j		
-	SupplementaryFig.6 SupplementaryFig.6a SupplementaryFig.6b	
Supl. 5a		
Supl. 5b		

Table: Summary of Changes To figures

Supl. 5c	SupplementaryFig.6c
Supl. 5d	SupplementaryFig.6d
Supl 5e	SupplementaryFig.6e
Supl 5f	SupplementaryFig.6f
Supl. 5g	SupplementaryFig.6g
	SupplementaryFig.7
Supl. 6a	SupplementaryFig.7a
Supl. 6b	SupplementaryFig.7b
Supl. 6c	SupplementaryFig.7c
Supl. 6d	SupplementaryFig.7d
	SupplementaryFig.7e New

Reviewer #1 (Remarks to the Author):

In this revised version, Dauben et al addressed my concerns, including the addition of a new main figure (Fig. 7) exploring the effects of PARG inhibition on ADPr dynamics in XRCC1 KO cells, as suggested. As a result, the revised manuscript is much improved and I recommend publication. However, some minor issues still require some attention:

Sup Fig. 2A: The chemical structure for adenosine is missing on the left side of the figure, causing the chemical structures for most molecules to be misaligned with their respective profiles/labelling.

Fig. 4C: the minus (-) and plus (+) signs are presumably swapped, as glucose starvation implies lack of glucose

We thank the reviewer for recommending acceptance of our manuscript. We are glad that we have addressed all their comments. We agree that adding the new main figure, focusing on PARG inhibition, as suggested by the reviewer, has improved the manuscript.

We have resolved the two minor issues identified in Sup Fig. 2A and Fig. 4C. Please note, that we stated “-H₂O” next to the adenosine structure in Sup Fig. 2A and in the table of Sup. Fig 2B, to display that we are searching for the mass of adenosine after a water loss, which is the standard approach in ADPr proteomics. We thank the reviewer for bringing these issues to our attention.

Reviewer #2 (Remarks to the Author):

The new experiments included in the revised manuscript showed some new biological insights using the developed antibody reagents. However, a few concerns still need to be addressed to demonstrate the superiority of these antibodies over other established ones.

We appreciate the reviewer's recognition that the new experiments included in the revised manuscript provide additional biological insight using the developed antibodies.

We note most of reviewer's remaining concerns relate to experiments and data that were already included in the original submission. Nevertheless, we address all points raised below, including those referring both to a newly added experiment and to original data.

1. Fig. 3B tried to evaluate affinity among various anti-poly-ADPr reagents for poly-ADPr-PARP1 at varied concentrations. The reported two synthetic antibodies were labeled with HRP molecules, but other commercial ones carrying Fc from different species would need the use of distinct secondary antibodies for detection. This experimental design cannot ensure fair comparison of their affinity and justify correlations of high signal intensities with sensitivities of primary antibody reagents.

We thank the reviewer for the opportunity to clarify the advantages of the modularity enabled by the SpyTag/SpyCatcher technology. Its application here to poly-ADPr detection—implemented for the first time—represents a key advance of our manuscript. This principle was previously illustrated experimentally for mono-ADPr in our 2023 *Molecular Cell* study (Longarini et al.) and is further supported by independent work (Hentrich et al., *Cell Chemical Biology*, 2021).

Importantly, the goal of Fig. 3B, which was already included in the original version of the manuscript, was not to compare affinities among different anti-poly-ADPr reagents, but rather to illustrate their practical performance in the formats in which the antibodies will ultimately be used in experiments. Consistent with this distinction, we use the term “sensitivity” for Fig. 3B, rather than “affinity”, which is used in Fig. 4 specifically in the context of affinity maturation of the mono-ADPr and site-specific antibodies.

By enabling rapid and controlled format switching, the SpyTag/SpyCatcher technology renders our modular ADPr antibodies fundamentally different from the Fc–WWE fusion reagent (MABE1031) and conventional IgG antibodies (CST and 10H), with distinct and unique advantages.

Detection of non-SpyTag primary antibodies typically relies on polyclonal secondary antibodies conjugated to a probe or enzyme such as HRP, thereby avoiding the laborious, unpredictable and expensive direct labeling of each primary antibody. In con-

trast, the SpyTag/SpyCatcher protein-engineering approach enables rapid, reproducible and covalent attachment of defined domains (e.g., Fc) or labels (e.g., HRP) directly to SpyTag-equipped antibody fragments, while preserving antigen-binding integrity.

Conventional antibody labeling most commonly relies on amine-reactive chemistries, such as N-hydroxysuccinimide esters, which are inherently non-site-specific and modify exposed lysine residues to variable extents, including those within or near the antigen-binding site. This leads to substantial batch-to-batch variability and carries the risk of impairing antibody function. By contrast, in our system labeling is performed exclusively on defined cysteine residues within the SpyCatcher component rather than on the antigen-binding moiety, thereby eliminating the risk of epitope disruption.

Importantly, chemical labeling of the SpyCatcher is performed independently and pre-labeled SpyCatcher reagents, including HRP-conjugated versions, are readily available as off-the-shelf SpyCatcher reagents. These can be directly ligated to SpyTag antibodies in a simple, one-step protein ligation reaction completed within ~1 h.

As a practical consequence of this, and particularly in the context of Fig. 3B, there is little rationale for using SpyTag antibodies—including the poly-ADPr antibodies developed here—in any detection format other than HRP–SpyCatcher for western blotting. Indeed, this distinction is already reflected in current practice: our previously published mono-ADPr antibody AbD43647 (Longarini et al. *Mol Cell* 2023) is always used for immunoblotting in the HRP–SpyCatcher format, whereas non-SpyTag reagents—including MABE1031, CST MAR/PAR and 10H—are employed using detection based on secondary antibodies.

Importantly, as we (Longarini et al., *Mol Cell* 2023, Fig. S1A) and others (Hentrich et al., *Cell Chem Biol* 2021, Fig. 3C) have shown, SpyCatcher–HRP–based detection provides higher sensitivity than both conventionally HRP-labeled IgGs and standard detection based on secondary antibodies. Extending the SpyTag technology to poly-ADPr antibodies therefore represents a key methodological advance of this manuscript, with clear practical advantages, as demonstrated in Fig. 3B.

Additionally, independent of the SpyTag technology and our antibodies, a direct comparison of binding affinities among the currently available reagents (MABE1031, CST MAR/PAR, and 10H) is inherently problematic. CST MAR/PAR and 10H are conventional bivalent IgG antibodies, whereas MABE1031 is an Fc–WWE fusion reagent that

is monovalent. The resulting avidity effects substantially enhance the apparent sensitivity of the bivalent IgGs relative to MABE1031, which lacks avidity, precluding a fair affinity comparison even among these existing tools. This principle is clearly illustrated by direct comparisons of monovalent and bivalent formats of the same antibody (Hentrich et al., *Cell Chem Biol* 2021). A truly fair comparison would require converting all reagents into the monovalent format; however, this is not possible due to the different nature of these tools.

Nevertheless, in line with the reviewer's suggestion, we performed a new experiment directly comparing the sensitivity of all antibodies in IgG or IgG-like formats. Our new data confirm that our previous observation of higher sensitivity of the HRP–SpyCatcher format (Longarini et al., *Mol Cell* 2023, Fig. S1A) also applies to the newly developed poly-ADPr antibodies. SpyTag antibodies are clearly less sensitive as synthetic IgGs (new Supplementary Fig 3B) a format that in practice is not used for immunoblotting. By contrast, the HRP–SpyCatcher format provides higher sensitivity within a simpler and faster workflow (no secondary antibodies required), as detailed above, and is the format in which our new poly-ADPr antibody AbD64138 is now available directly as an off-the-shelf reagent through Bio-Rad (Product Code TZA0117P).

2. In Fig. 3A-E, 4F, and 7A, authors claimed significantly higher signals from the reported antibodies. To support these claims, quantitative analysis of the immunoblots with at least three independent replicates would be required.

We thank the reviewer for raising this point. The immunoblot experiments in Figs. 3A–E, 4F, and 7A, as presented in the original manuscript, were intended to assess relative detection sensitivity under identical experimental conditions, rather than to provide quantitative measurements of ADPr levels. As such, these blots are shown as representative examples from three independent experiments that yielded consistent results, in line with common practice for immunoblot-based sensitivity comparisons.

We note that quantitative densitometry is not well suited for comparing reagents with distinct formats, valencies, and detection chemistries, as signal intensity does not scale linearly with binding affinity or sensitivity. Nevertheless, to further support our conclusions, we have now added quantification of the Western blots in Supplementary Figures 3-5 and 7 using a consistent, ROI-based analysis with uniform background

correction. However, the comparison is technically constrained by the markedly different signal ranges of the compared antibodies, ranging from close to detection limit to the upper dynamic range of ECL detection before saturation. As a result, quantitative interpretation, especially for low intensity bands, is sensitive to noise and background estimation. All images used for quantification were non-saturated, but due to the limited overlap in the linear dynamic range of the compared antibodies, a single exposure allowing fully linear quantification of both signals was not achievable. For this reason, we present the quantification as a supportive information in the supplementary figures and rely primarily on the representative blots shown in the main figures, including multiple exposures, to illustrate the substantial differences in antibody performance.

3. In Fig. 4C, the authors claimed that AbD41122 allows to detect mono-ADPr on RNA molecules. How does this affinity-matured antibody compare with other commercial ones like CST D9P7Z, MABE1016, and MABE1076 for recognition of mono-ADP ribosylated RNA?

We thank the reviewer for this question. In Fig. 4C, we compared the affinity-matured mono-ADPr antibody AbD41122 to AbD43647, our previously published mono-ADPr-specific antibody (Longarini et al., *Mol Cell* 2023), which is widely used in the field and is generally regarded as a gold-standard reagent for specific mono-ADPr detection.

While the original version of the manuscript already demonstrated that AbD41122 displays binding preferences complementary to those of AbD43647, as detailed in our previous point-by-point response, the experiment shown in Fig. 4C was specifically performed to address the reviewer's earlier concern that the advantage of AbD41122 was not evident, given its similar sensitivity to AbD43647 for protein mono-ADPr, although with a different recognition pattern. We have addressed this concern by including in the manuscript data from a separate collaborative project on RNA ADPr. As shown in Fig. 4C, AbD41122 performs significantly better than AbD43647 in detecting RNA mono-ADPr.

By contrast, CST D9P7Z and MABE1016 are pan-ADPr antibodies with broad reactivity toward both mono- and poly-ADP-ribosylation, making direct comparison with mono-ADPr specific antibody less informative. This distinction represents a central conceptual point of our study, which aims to move beyond pan-ADPr detection by providing reagents with defined mono- or poly-ADPr specificity, as already discussed

in the manuscript. Moreover, the RNA analyzed in Fig. 4C is derived from a cellular context in which multiple ADPr species may be present on RNA, further limiting the interpretability of signals obtained with pan-ADPr antibodies with respect to mono-ADPr-specific modification.

Nevertheless, in response to the reviewer's point, we performed additional comparative experiments (new Supplementary Fig. 5C and below figure) to further benchmark AbD41122 against all the commercially available reagents suggested by the reviewer. MABE1076 is a mono-ADP-ribose binding reagent, making it a more conceptually relevant comparator. However, while both AbD43647 and AbD41122 detects starvation-induced ADPr on RNA, as shown in Fig. 4C, no increase in RNA ADPr compared to control was observed with either MABE1076 or MABE1016, as shown in the below figure. This indicates that, for these two reagents, the detected signal is either entirely nonspecific or so dominant that it masks any specific signal. Consistently, whereas the AbD41122 signal is abolished by RNase treatment (Fig. 4C), MABE1076 and MABE1016 remain largely unaffected (see figure below), confirming their nonspecific binding and indicating that a component other than RNA underlies their nonspecific signal. Therefore, these two reagents cannot be used to study starvation-induced ADPr on RNA.

By contrast and consistent with a previous report (Weixler et al., *Life Science Alliance*), the poly/mono-ADP Ribose CST D9P7Z antibody produced a strong specific signal (new Supplementary Fig. 5C). However, CST D9P7Z, is a pan-ADPr antibody recognizing both poly- and mono-ADPr and its signal therefore reflects the combined detection of multiple ADPr species and therefore lacks mono-ADPr specificity. Notably, CST D9P7Z exhibits particularly strong recognition of poly-ADPr — stronger than other pan-ADPr antibodies, including MABE1016 and our previously described AbD33641 (Bonfiglio et al. *Cell* 2020). As a consequence, the CST D9P7Z signal can be dominated by poly-ADPr that is continuously formed in cells even in the absence of exogenous DNA damage, thereby masking the detection of even strongly induced mono-ADPr, such as that induced by interferon treatment, as briefly discussed in the original version of our manuscript and experimentally shown by Kar et al. *EMBO J.* 2025. Accordingly, the strong CST D9P7Z signal reflects the combined recognition of multiple ADPr species, limiting its interpretability in direct comparison with the mono-ADPr-specific antibody AbD41122.

Therefore, these new results, obtained by precisely following the reviewer’s request, confirm our original conclusion that AbD41122 is the best mono-ADPr-specific antibody for RNA ADPr, at least in the context of starvation-induced RNA ADPr, an established condition for this modification (Weixler et al., *Life Science Alliance*).

4. Lines 238 and 307, olaparib can inhibit PARP1 and other PARPs.

We thank the reviewer for pointing this out. Although we did not refer to olaparib as a PARP1-specific inhibitor, but rather as a “PARP1 inhibitor,” in contrast to AZD9574, which we describe as a “PARP1-specific inhibitor,” we agree that the wording could be confusing in this context. We have therefore revised the text to explicitly refer to olaparib as a “PARP inhibitor” and to clearly distinguish it from the PARP1-specific inhibitor AZD9574.

5. The glucose label for Fig. 4C is confusing. Should be glucose starvation.

We agree with the reviewer that the glucose label for Fig. 4C is confusing. As also noticed by Reviewer #1, the minus (-) and plus (+) signs should have been swapped. We have corrected this mistake and thank the reviewer for pointing it out.

Reviewer #3 (Remarks to the Author):

I thank the Authors for addressing all my comments, as well as many of those raised by other reviewers.

I particularly appreciate the new experiments exploring the specificity and sensitivity of the anti-poly antibodies, which I do think will make very useful tools. I now understand the challenges with quantifying the 'punctate' nature of the observed ADPr signals in cells, and with testing branchpoints, and I thank the Authors for explaining those. I also understand the challenge with Ponceau stain brightness, when little total protein is loaded; this is a common technical limitation.

Given the fact that, to me, the main value of the article lies in its methodological aspects (both in terms of antibodies and MS insights), I wonder whether the authors could still consider a different article title - one that does not highlight primarily the, somewhat secondary, biological insights, but rather both the antibodies and MS insights, presenting these as tools for monitoring PAR and MAR in different contexts. But this is just a gentle suggestion.

Overall, I think this manuscript is ready for publication and will be useful to the community. Congratulations.

We are pleased that the reviewer's concerns have been addressed by our revisions. We thank the reviewer for the positive assessment that the manuscript "will be useful to the community" and agree that the anti-poly antibodies "will make very useful tools".

In response to reviewer suggestions emphasizing the methodological value of the work, we have revised the title to better reflect the central contributions of the methodological advances, with biological applications presented as proof-of-principle. Specifically, we have changed the title from "A Versatile Toolbox Reveals XRCC1 Deficiency-Driven Reshaping of the PARP1 Mono-ADP-Ribosylation Wave" to "Versatile and sensitive detection of mono- and poly(ADP-ribosyl)ation reveals XRCC1-dependent remodeling of PARP1 signaling".